# The Impact of Additive Manufacturing on Supply Chain Management from a System Dynamics Model—Scenario: Traditional, Centralized, and Distributed Supply Chain

Jairo Nuñez Rodriguez [1,*], Hugo Hernando Andrade Sosa [2], Sylvia Maria Villarreal-Archila [3] and Angel Ortiz [4]

1 Faculty of Industrial Engineering, Universidad Pontificia Bolivariana, Km 7 Vía Piedecuesta, Piedecuesta 681017, Colombia
2 Computer Systems Engineering, Research Group SIMO, Universidad Industrial de Santander, Calle 9 con Carrera 27, Bucaramanga 680006, Colombia
3 Faculty of Industrial Engineering, SOLYDO Researcg Group, Unidades Tecnológicas de Santander, Av los Estudiantes 982, Bucaramanga 680005, Colombia
4 Research Centre on Production Management and Engineering, Universitat Politècnica de València, Camino de Vera s/n, 46022 Valencia, Spain
* Correspondence: jairo.nunez@upb.edu.co; Tel.: +57-6796220

**Abstract:** In order to describe the impact that the appropriation of additive manufacturing (AM) has on the supply chain (SC), a validated system dynamics model representing vectorially multiple products and multiple demands in different periods was used as a basis to apply to a case study of medical implant manufacturing, configuring three chain scenarios: 1. traditional supply chain with subtractive manufacturing, 2. centralized supply chain with additive manufacturing, and 3. decentralized supply chain with additive manufacturing. It was possible to notice that the production time is longer in additive manufacturing compared to traditional manufacturing and the cycle time and total demand closure were lower in traditional manufacturing. In addition, it was observed that the AM performance is significantly better in conditions of lower demand, which can be attributed to the characteristics of customization and small batches that this type of production approach implies.

**Keywords:** supply chain (SC); additive manufacturing (AM); system dynamics (SD); healthcare sector

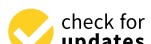



## 1. Introduction

Additive manufacturing (AM) is one of the disruptive technologies that can alter the behavior of industries. Regarding the supply chain, it was mentioned that it reconfigures its management and the processes it includes. Thus, based on a system dynamics model proposed by Nuñez-Rodriguez et al. [1], which presents the operation of a supply chain composed of three links, supplier, manufacturer, and distributor, with a make-to-order (MTO) system, three application scenarios are designed where it is possible to visualize the behavior of the MTO chain with subtractive and additive characteristics in terms of material and information flows through cycle times, quantities of raw material, and quantities of the finished product.

The system dynamics representation of the model proposes the behavior of a multi-demand, multi-product motor company [2], with variable capacity and different characteristics for each product. It includes suppliers of a single raw material, corresponding to an easily accessible commodity product that does not require a specialized supplier, and distributors with varying delivery times depending on the selected mode of transportation. These actors are articulated in the supply chain that functions as a collaborative system to respond to demand. This behavior can also be seen in a traditional supply chain, thus facilitating the contrast and quantification of the impact of the manufacturing approach in the chain's management.

For the simulation of the scenarios, the starting point is the healthcare sector and its performance in Colombia. Possible changes in supply chain management were analyzed based on the inclusion of additive manufacturing, particularly in the case study of programmed surgery. The initial data were provided by the Interfaz Research Group of the Universidad Industrial de Santander, Colombia.

The scenarios presented represent two alternatives for the appropriation of the additive process, depending on the role played by the manufacturer and how it interacts with the other actors in the chain, given that it can be centralized, i.e., one manufacturer serving three regions, or decentralized, where three focal manufacturers each serve their own region. According to the description of the scenario, three regions were projected; in this case, the central region (R1), the coastal region (R2), and the southern region (R3), as shown in Figure 1, assuming the Colombian geography.

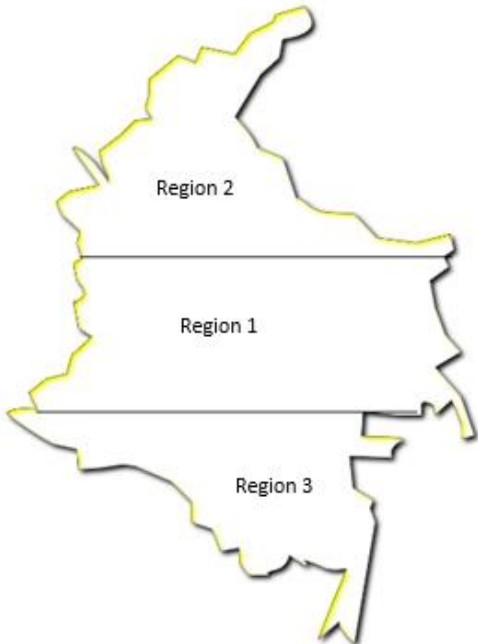

**Figure 1.** Regions proposed for model and scenario simulation. Source: own elaboration.

According to the information provided, it was estimated that by 2020 there were 890 cases related to the use of medical devices in Colombia per year. These devices are now divided into three categories: implants, biomodels, and cutting guides, as shown in Figure 2.

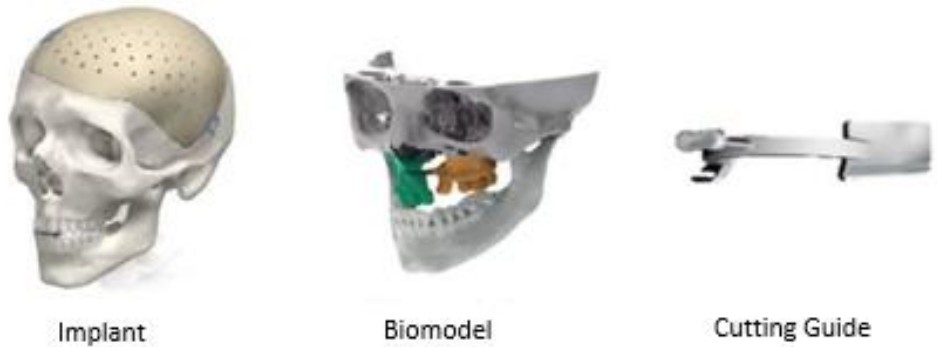

**Figure 2.** Categories of specific medical devices. Source: Interfaz Research Group and Grice, 2020.

The demand and printing times were projected for the three product categories previously described. After analyzing the presented cases, it was possible to notice that 60% of the cases were concentrated in region 1 (central), 30% in region 2 (north), and the re-

maining 10% in region 3 (south). Products were classified into three categories: product 1 corresponds to biomodels, product 2 to cutting guides, and product 3 to implants. Table 1 breaks down the projected product quantities by region and product.

**Table 1.** Projected annual and monthly demand for medical devices.

| Aggregate Demand | Region 1 | | Region 2 | | | Region 3 | Annual Total |
|---|---|---|---|---|---|---|---|
| | Annual | Monthly | Annual | Monthly | Annual | Monthly | |
| Total Product 1: Biomodel | 534 | 45 | 267 | 22 | 89 | 7 | 890 |
| Total Product 2: Cutting Guide | 224 | 19 | 112 | 9 | 37 | 3 | 373 |
| Total Product 3: Implant | 101 | 8 | 51 | 4 | 17 | 1 | 169 |
| Total | 859 | | 430 | | 143 | | |

Source: own elaboration based on data provided by Interfaz Research Group.

Based on the context, it was possible to carry out the following actions: simulation of the scenarios, results comparison, and analysis of the impact of the appropriation of additive manufacturing on the supply chain.

## 2. Literature Review

The healthcare supply chain (HSC) is held as a value chain network where companies dedicated to the development of care package elements (raw material suppliers and manufacturers) and those dedicated to the provision of care services (provider entities) interact to meet the need for the medical care of patients (consumers) [3]. Due to the high complexity of manufacturing and distribution of this type of goods and services, some optimization processes are required to integrate the supply chain with the management of material, information, and financial flows [4,5] since, under the current conditions, the direct transfer of additive manufacturing knowledge from the industrial sector (suppliers) to the care sector is complex [6].

From a macro perspective, the HSC should be analyzed based on the information, supplies, and finances involved in the acquisition and transfer of goods and services from the supplier to the end user in order to improve clinical outcomes while controlling costs; this also includes the identification of crucial actors and the role they play within the chain [3]. In this sector, the role of the state is fundamental since it is accountable for guaranteeing access to healthcare and determining and directing the entities in charge of administering the services [7].

For the definition of scenarios, the results of a systematic literature review were considered, consisting of application cases within different medical procedures. Table 2, referenced in Appendix A, focuses on the results from the components of the AM process, starting with the inputs that include the types of materials explored to produce medical devices or biomedical equipment. After that, the types of 3D printing and the cases in which they have been applied are shown. Finally, the outputs related to the cases of medical procedures are summarized and then classified according to the service time.

Among the materials, five types are highlighted: titanium, polymers, ceramics, metals, and nanometals, with titanium being the most used for medical devices such as surgical instrumentation or surgical kits [8]. In addition, it is paramount to take into account the purpose of the element to be produced, since there are tolerance levels for its use within the product flow; for example, when implants or invasive components are made, the patient's body may or may not accept this material.

The transformation processes correspond to the existing types of 3D printing, which made it necessary to consider the variables of printing time, implementation costs, ease of usability, access to preventive and corrective maintenance, and others [9] to select the most appropriate one. The most representative cases where these processes are used are aneurysm, cancer, cardiovascular disease, skull surgery, surgical guides, maxillofacial surgery [10], dentistry [11,12], orthopedics [13], prosthetics [14], vascular surgery [15] and others. In addition, these cases are characterized by the time availability when planning

and waiting for materials. The above suggests that, currently, AM is not as responsive as traditional manufacturing in medical emergency cases, considering the high levels of customization and time needed for printing procedures and finishing processes.

**Table 2.** Additive manufacturing process in the health sector.

| Application Cases | |
|---|---|
| **INPUTS** | **Materials**: (Sidambe, 2014), (Bose et al., 2018). <br> **Titanium**: (Abe et al., 2003), (Leuders et al., 2013), (Yves-Christian, H., Jan, W., Wilhelm, M., Konrad, W., & Reinhart, 2010), (Hrabe & Quinn, 2013a), (Hrabe & Quinn, 2013b), (Sahoo, 2014), (El-Hajje et al., 2014), (Beaucamp et al., 2015), (Elahinia et al., 2016), (Dadbakhsh et al., 2016), (Wang et al., 2016), (Zhai et al., 2016), (Sahoo & Chou, 2016), (Hinderdael et al., 2017), (MacBarb et al., 2017), (Fatemi et al., 2017). <br> **Polymers**: (Cruz & Coole, 2006a), (Lopes, G., Miranda, R. M., Quintino, L., Rodrigues, 2007), (Tröger et al., 2008), (Höfer & Hinrichs, 2009), (Suwanprateeb et al., 2014), (Husár et al., 2014), (Short et al., 2014), (W. Z. Wu et al., 2014), (Vaezi & Yang, 2015), (Leonards et al., 2015), (Poh et al., 2016), (Jungst et al., 2016), (Stieghorst et al., 2016), (Pan et al., 2017), (Walker et al., 2017), (Pekkanen et al., 2017), (Shin et al., 2017), (Liravi & Toyserkani, 2018), (Kuo et al., 2018). <br> **Ceramics**: (Cruz & Coole, 2006), (Yves-Christian et al., 2010), (Goffard & Sforza, 2013), (Lusquiños et al., 2014), (Gmeiner & Deisinger, 2015), (Falvo D'Urso Labate et al., 2017), (Nabiyouni et al., 2018), (Choi et al., 2018). <br> **Metals**: (Srivatsan & Sudarshan, 2015), (Hong et al., 2016). <br> **Nanomaterials**: (Dobrzański, 2007), (Sugioka & Cheng, 2014), (Kong et al., 2016), (Ramasamy & Varadan, 2016), (Koumoulos et al., 2017), (BRUBAKER et al., 2017), (Ji et al., 2017), (Misra et al., 2017a). |
| **TRANSFORMATION** | **Stereolithography**: (Melchels et al., 2010), (Cooke et al., 2003), (Gauvin et al., 2012), (Melchels et al., 2009), (Dhariwala et al., 2004), (Bill et al., 1995), (D'Urso et al., 2000), (Lee et al., 2007). <br> **Fused deposition modeling (FDM)**: (Zein et al., 2002), (Schantz et al., 2005), (McCullough & Yadavalli, 2013), (Mohamed et al., 2015), (Espalin et al., 2010), (Gronet et al., 2003), (Xu et al., 2014). <br> **Selective laser sintering (SLS)**: (Rogers et al., 2007), (Clinkenbeard et al., 2002), (Berry et al., 1997), (Schmidt et al., 2007), (Rimell & Marquis, 2000), (Shishkovsky et al., 2008), (Williams et al., 2005), (Edith Wiria, Sudarmadji, et al., 2010), (Edith Wiria, Fai Leong, et al., 2010), (Kruth et al., 2003), (Duan & Wang, 2011). <br> **Selective laser melting (SLM)**: (Vandenbroucke & Kruth, 2007), (Strano et al., 2013), (Attar et al., 2014), (Chlebus et al., 2011), (Mullen et al., 2009), (Zhang et al., 2011), (Wei et al., 2015), (Yang et al., 2012). <br> **Electron beam melting**: (Facchini et al., 2009), (Cronskär et al., 2013), (Ramakrishnaiah et al., 2017), (Koptioug et al., 2012), (Murr et al., 2011), (Li et al., 2009), (Murr et al., 2012), (Koike et al., 2011). |
| **OUTPUTS** | **Aneurysm**: (Opolski et al., 2014), (Ho et al., 2017), (Ryan et al., 2016). <br> **Cancer**: (Petcu, 2017), (Witowski, Pędziwiatr, et al., 2017), (Gallivanone et al., 2016). <br> **Cardiovascular**: (Nocerino et al., 2016), (Kuk et al., 2017), (Misra et al., 2017b), (Lueders et al., 2014), (Arcaute & Wicker, 2008), (Smith et al., 2017), (Cheng & Chen, 2006). <br> **Skull**: (Berretta et al., 2018), (Jardini et al., 2014), (Peel et al., 2017), (Winder et al., 1999), (Msallem et al., 2017). <br> **Surgical guides**: (Popescu et al., 2015), (Bibb et al., 2009), (Dahake et al., 2017), (Dahake et al., 2016). <br> **Maxillofacial**: (Thomas et al., 2014), (Daniel & Eggbeer, 2016), (Singare et al., 2006), (Sljivic et al., 2016), (W. Wu et al., 2010), (Al-Ahmari et al., 2015), (Winder & Bibb, 2005), (Brito et al., 2016). <br> **Dentistry**: (Gebhardt et al., 2010), (Budzik et al., 2016), (Jiménez et al., 2015), (Nayar et al., 2015), (Liu et al., 2006), (Faure et al., 2012). <br> **Orthopedic**: (Sankar et al., 2017), (Jackson et al., 2017), (Wong, 2016), (Sindhu & Soundarapandian, 2017), (Popovich et al., 2016), (M Zanetti et al., 2017), (Popescu et al., 2017), (Chougule et al., 2014), (Nakano & Ishimoto, 2015), (Li et al., 2017), (Blaya et al., 2017), (de Beer & van der Merwe, 2013), (Huang et al., 2015), (Ahn et al., 2006), (Tie et al., 2006), (Ogden et al., 2014). <br> **Prosthesis**: (Lathers & La Belle, 2016), (Rahmati et al., 2012), (Radosh et al., 2017), (Hagedorn-Hansen et al., 2016), (Zuniga et al., 2015), (Schrank et al., 2013), (Vitali et al., 2017). <br> **Vascular**: (O'Hara et al., 2016), (Ionita et al., 2014), (Spallek & Krause, 2016). <br> **Others**: Liver surgery (Witowski, Coles-Black, et al., 2017), (Soon et al., 2016). Plastic surgery: (Bauermeister et al., 2016). |

Source: Own elaboration based on systematic literature review. References are available in Appendix A.

Based on the cases mentioned above, contrasted with the characterization of the SSC in Colombia (subtractive manufacturing approach), a proposal was made on how AM would impact the chain management. The four levels of appropriation of the additive process proposed by J. Chen et al. (2016) [16] were taken as a reference: (1) printing for surgical procedure planning, (2) printing of operation tools through implant guides, (3) artificial bone printing, and (4) organ printing.

Consequently, the first representative change would be evident in the chain structure, where the roles of some actors could change and, in addition, new categories related to the design activity could appear, considering that, in AM, the collection of personalized patient information is necessary. Based on this information, the three scenarios analyzed were defined.

From a system dynamics approach, 306 cases that related supply chain management to system dynamics were reviewed. Researchers found that 75% of the results are linked to analyzing the management process variables from the supply chain, where one or two variables that model a specific process are studied. In most of the application cases, a specific issue of a variable affecting a management process is modeled, i.e., the aim is to understand the behavior of a variable within the performance of the chain. The predominant variables tend to be inventory management, information exchange, chain integration, system performance, demand projection and management, and production planning and scheduling. However, no studies relate all the chain management processes together from a holistic viewpoint; although the chain is considered a system where the associated variables affect others, there is not any research in the literature that integrates the eight SC management processes into a single model.

In some cases, the SD approach was oriented to determine the impact of demand variability and delivery time on SC performance; in other cases, models were proposed to simulate the operation of chain networks, focusing on the definition of metric indicators such as inventory, orders, and customer satisfaction. Likewise, multi-objective analyses were developed for policy formulation, modeling supply and demand in single markets. Another application focus studied the behavior and relationships within the chains to determine the critical components that interfere with efficiency and sustainability results.

In contrast to the analysis of the evolution of SD models related to the SC, there were evident coincidences among the variables analyzed in chronological order (demand analysis, inventory analysis, bullwhip effect, costs, production capacity, distribution, time), i.e., system dynamics has focused on the analysis of traditional industry variables. The latter becomes an opportunity for the development of simulation models that allow analyzing the behavior of the chain and its management based on the variables that industry 4.0 supposes, integrating trends such as additive manufacturing, big data, augmented reality, industrial internet of things, and cloud storage, among others, which have been characterized as potential agents of change in the configuration and behavior of the industry.

## 3. Materials and Methods

System dynamics is a modeling process for the understanding and discussion of complex problems such as supply chains where a series of actors interact in response to flows of money, information, and materials [2]. Starting from the base model proposed by Nuñez-Rodriguez et al. [1], built with the guidance of Andrade [17], and validated with Sterman's process stages [18], the behavior of cycle times, raw material quantities, and finished product quantities were analyzed in contrast to scenarios with subtractive and additive characteristics. The results were obtained with Evolution software, version 4.6, created by Research Group SIMON, Universidad Industrial de Santander, Bucaramanga, Colombia in 2003.

Figure 3 presents the flow-level diagram of the model's behavior that recreates a multi-product supply chain, considering the definition of the order acceptance policies and the purchasing policy. This representation corresponds to a simplified version of the supply chain and order management, developed in three processes: purchasing (supplier),

production (manufacturer), and distribution (distributor). This model operates in a vectorial way, except for the supplier link, and allows the representation of the behavior through the supply chain with three types of products, 1(X), 2(Y), and 3(Z), with different material consumption, processing times, and distribution times for each one.

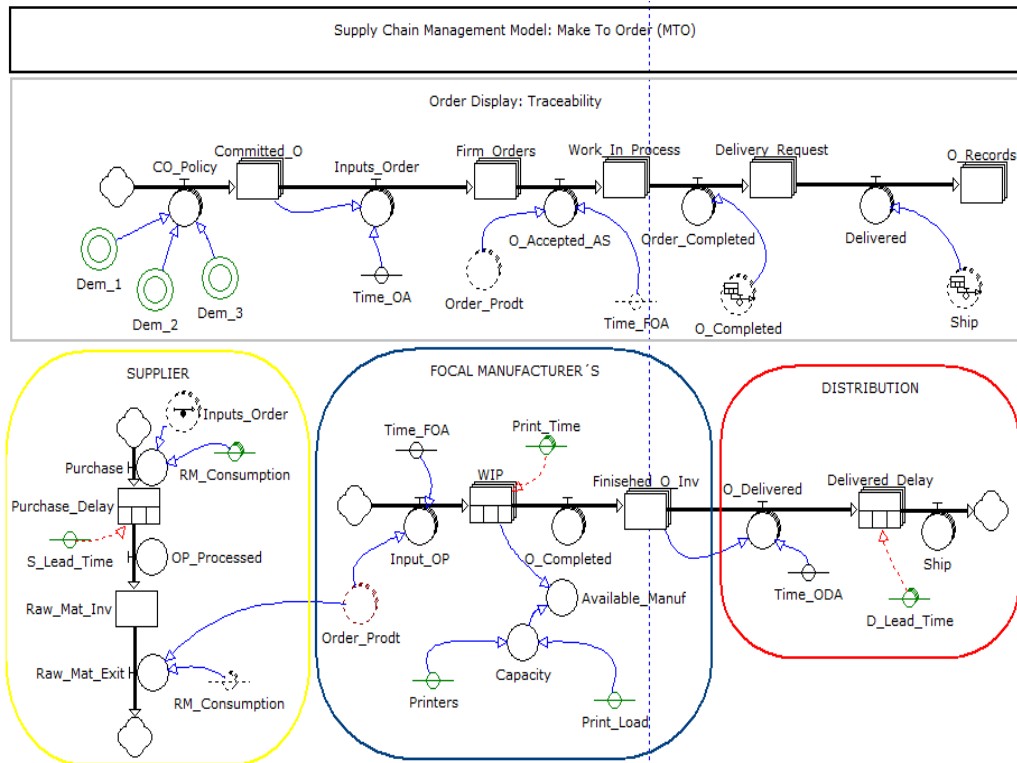

**Figure 3.** Flow and level diagram. Supply chain model, based on orders (MTO). Reprinted with permission from Nuñez-Rodriguez, et al. [1]; published by Processes, 2021.

The model has structural operating policies associated with stock acceptance and capacity acceptance, as shown in Figure 4. These are policies that the focal manufacturer considers to prioritize order processing and the generation of production orders.

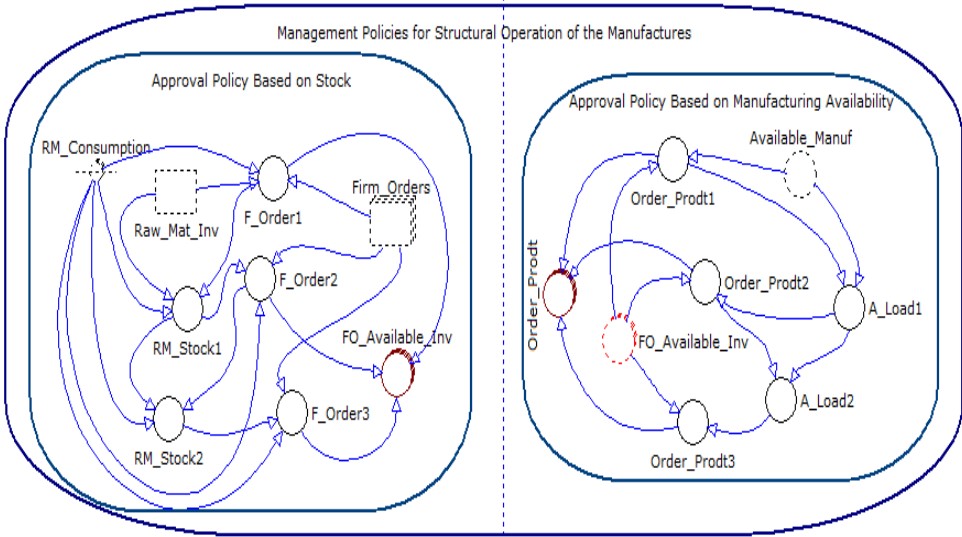

**Figure 4.** Structural performance policies of the model. Reprinted with permission from Nuñez-Rodriguez, et al. [1]; published by Processes, 2021.

Some variations were made in the interest parameters to reflect the changes implied by the AM within the chain, starting with the inclusion of additive process activities such as the design of 3D files, printing, and finishing processes, which, although not deepened in the model, are assigned to the manufacturer. Given the lack of data in practice due to the appropriation level of AM in the production area, a traditional behavior and two additive behaviors were proposed for a simplified supply chain with data assumed from traditional cases documented in the literature.

The scenario approach was carried out for the health sector case study, specifically for medical implant manufacturing. Through a literature review, it was possible to determine some adjustments which allowed the representation of two alternatives for the appropriation of the additive process, depending on the role played by the manufacturer and how the interaction with the other actors in the chain takes place. In summary, three scenarios were constructed: the first represents a traditional supply chain (TSC) under the original conditions of the conceptual model, the second represents a centralized additive supply chain (ASC), and the third represents a decentralized ASC. Finally, the results obtained in each scenario were contrasted to determine the impact of the additive process on the supply chain performance.

## 4. Results

### 4.1. Description of Scenarios

#### 4.1.1. Scenario 1—Traditional Supply Chain

As for the traditional supply chain scenario, the use of subtractive manufacturing processes, such as the generation of products from the removal of material [19], includes the following: manual extraction activities, traditional machining or CNC machining [20], as well as complementary post-processes to obtain medical devices, dental devices, and implants [21]. Regarding supply chain management, the structure can be decentralized or centralized, where a high level of coordination and integration is required in developing, manufacturing, and delivering inputs related to the elements of care packages (classified into medical devices, drugs, and biotechnologies).

In this sense, the expected behavior is that the production processes are carried out by the first link, i.e., the supplier, who supplies the healthcare provider institutions [3].

The TSC representation is shown in Figure 5, with three service-provider regions (1, 2, 3) that produce a specific product (1, 2, 3), which can be consumed in any of the studied regions with an associated distribution transport time. In this case, there is a raw material supplier, a focal manufacturer, and a distributing company in each supply chain. The model allows different operating policies to be established.

In order to simulate the scenario, it was necessary to modify the base model [1]. First, researchers began by representing a priority of care policy by region and adding the demand policy, as shown in Figure 6.

This policy defines the priority of attention by region based on product demand (Dem_Pi_Ri). Initially, the priority is to deliver the product in its own region, and then continue with the closest regions (P_A_D_MRi). This policy can be modified according to the conditions of the problem to be modeled. The main characteristic to be considered in the modeling is the consistency of the product delivery times, i.e., delivery to the closest region should not take longer than farther regions. The parameters and variables considered are described below:

P_A_D_MRi: parameter that determines the priority of attention order for regions 1, 2, and 3 by the company MOTORAi_Ri (i: 1,2,3).

Dem_Pi_Ri: exogenous variable determining the product demand by region in a given period (i: 1,2,3).

T_Dist_P_A_MRi: shows the parameter to keep in mind as, if the priority order is modified, the distribution times must be adjusted coherently for each region (i) and the elements of the two vectors must be in the same order.

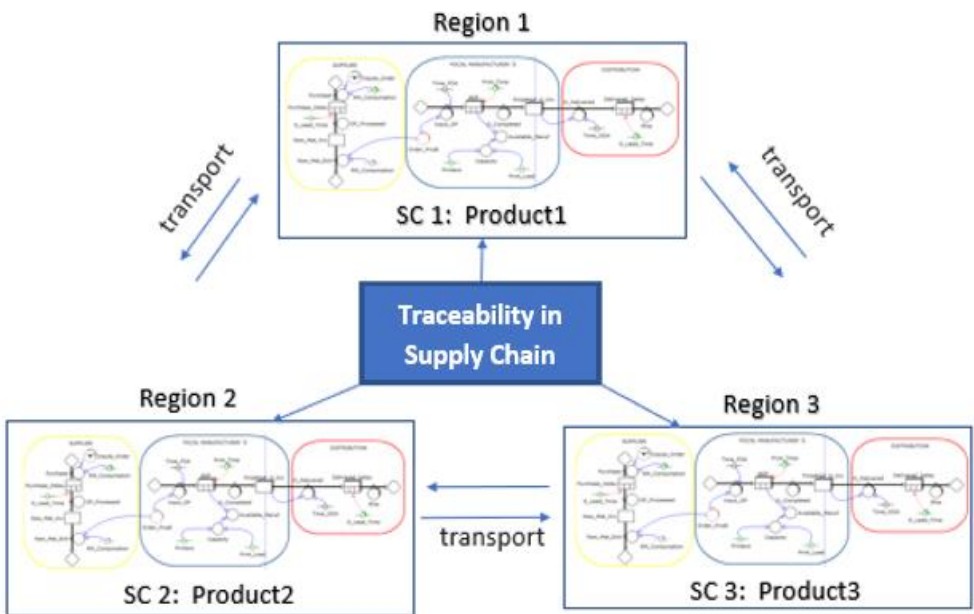

**Figure 5.** Scenario 1 representation—traditional supply chain. Source: own elaboration.

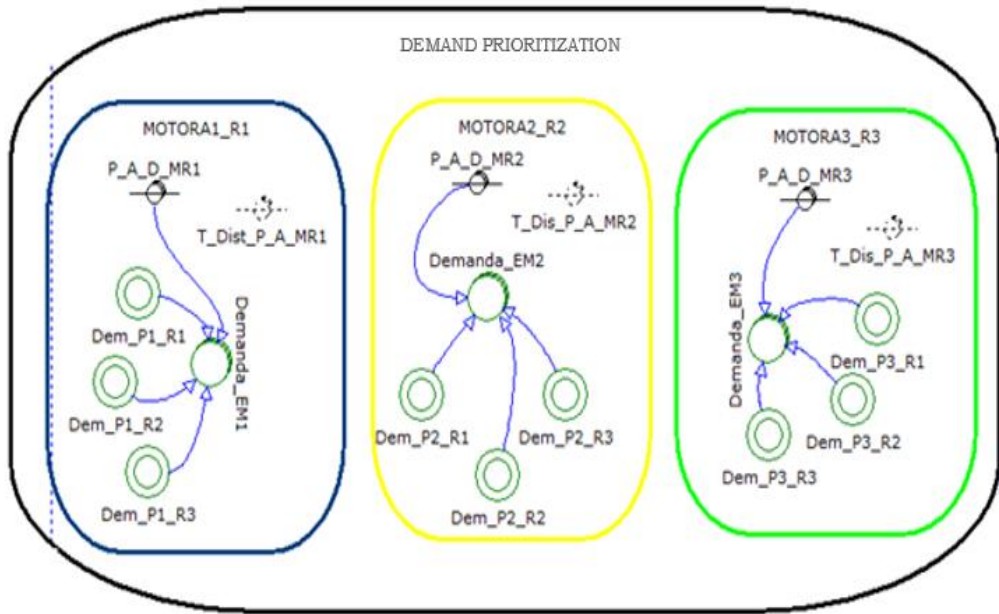

**Figure 6.** Demand priority policy—scenario 1: own elaboration. Evolution software.

Secondly, the supply chain traceability was visualized, consolidating the production order status information as shown in Figure 7, where the entire system can also be seen. With the consolidated modeling, nine vectors are created to operate each layered level. The first three vectors are ordered according to the attention priority policy of the region. The demand of the first manufacturer (Demand_FM1) corresponds to what supply chain 1 must produce. Each vector represents each region's demand; the same happens with the remaining two supply chains (2 and 3). Thus, it continues with the creation of production orders and the output of products in the process (outputWIP1) and shipped products (Shipped1).

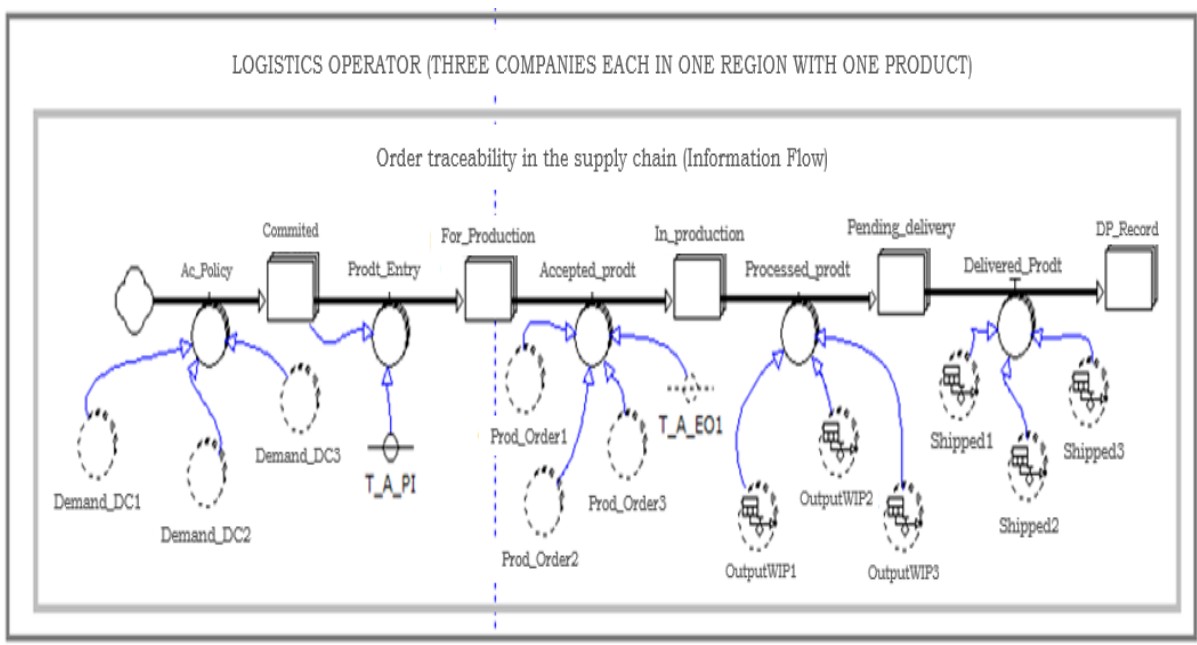

**Figure 7.** SC traceability—scenario 1: own elaboration. Evolution software.

Thirdly, the SC of region 1 produces a single product and meets the demand of the three regions. The supplier is maintained with a single raw material as in the base model; consumption is constant because it is the same product (Raw_Mat_Quant), as seen in the supplier link in Figure 8, which is repeated in each of the regions.

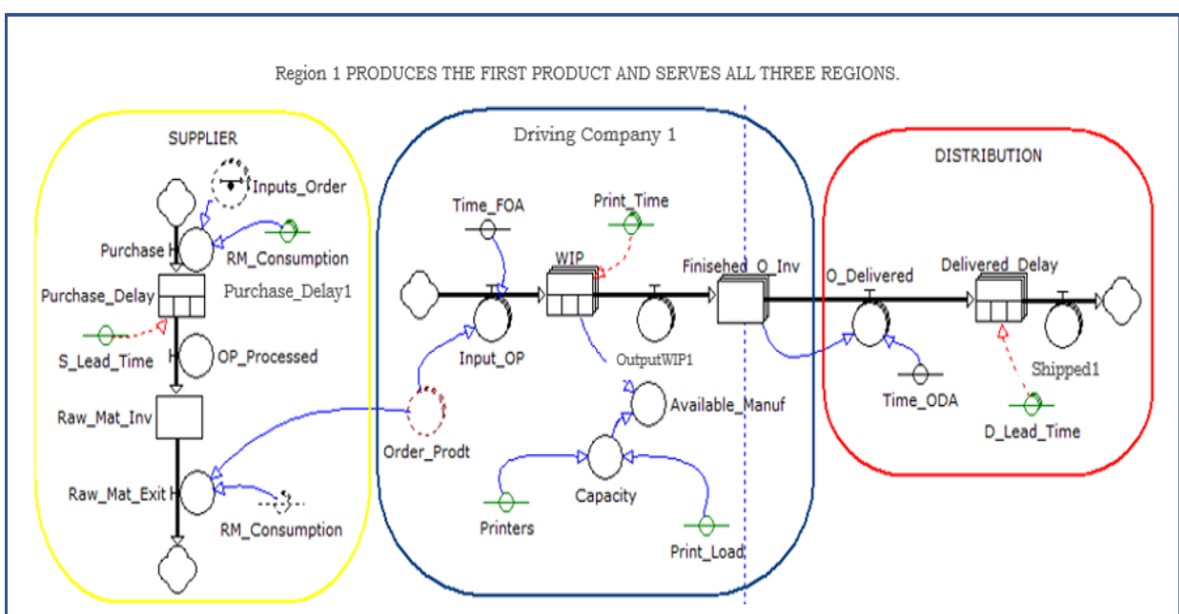

**Figure 8.** Operation in region 1—scenario 1: own elaboration. Evolution software.

On the other hand, focal manufacturer 1 may have different capacity characteristics (Total_Load_Cap1) concerning the focal manufacturers in regions two and three. This situation is replicated with the processing time (Printing_T1) since one product is produced per region, and the characteristics vary from one area to another. In this case, the management policy for the structural operation of the first manufacturer, Figure 4, is involved; the latter changes when it comes to the base model since the priority is given to the region to be served and not to the product (the order of arrival). Distribution time is considered a vital

aspect of this model because it can represent asymmetry in shipments between regions, affecting production delivery times and the cycle time to meet demand.

In order to contrast the models and determine the impact of AM in terms of chain cycle time, researchers retained the annual and monthly demand distribution data for the scenario 1 simulation. On the other hand, since each region requires the three products regardless of where they are produced, considering the layers by vectors explained above, Table 3 shows the configuration of products by the manufacturer. The model allows modifying the discussed policy definition [1].

**Table 3.** Vector organization of orders by manufacturers.

| V1 | V2 | V3 | V4 | V5 | V6 | V7 | V8 | V9 |
|----|----|----|----|----|----|----|----|----|
| | FM 1 | | | FM 2 | | | FM 3 | |
| P1R1 | P1R2 | P1R3 | P2R1 | P2R2 | P2R3 | P3R1 | P3R2 | P3R3 |

Source: own elaboration.

The capacity, materials, and time conditions were defined for each of the three focal manufacturers, as shown in Table 4. The grams consumed by each product, the number of machines, and the capacity per machine are related. In addition, the procurement times for each region, the manufacturing/printing times associated with the product size, and the distribution times according to proximity are defined. For example, when delivery occurs in the same region, the estimated time is 24 h, for the closest area it is 48 h, and for the farthest region, it would take 72 h to deliver the final product.

**Table 4.** Scenario 1 conditions.

| | | FM 1 | FM 2 | FM 3 |
|---|---|---|---|---|
| Supplier | Material consumption | Grams | Grams | Grams |
| | Product 1 | 81 | | |
| | Product 2 | | 14 | |
| | Product 3 | | | 21 |
| | Lead-time supplier–supplier | 120 | 120 | 120 |
| Focal Manufacturing | Production conditions | Units | Units | Units |
| | Number of machines | 4 | 4 | 4 |
| | Load per machine | 0.25% | 0.25% | 0.25% |
| | Processing time | | | |
| | Product 1 | 12 | | |
| | Product 2 | | 4 | |
| | Product 3 | | | 5 |
| Distribution | Distribution time | Region 1 | Region 2 | Region 3 |
| | Region 1—central | 24 | 48 | 48 |
| | Region 2—north | 48 | 24 | 72 |
| | Region 3—south | 48 | 72 | 24 |

Source: own elaboration.

In this scenario, the order priority policy is set by region, as summarized in Table 5, where, firstly, each manufacturer meets the demand of the product it manufactures. For example, in region 1, focal manufacturer 1 first finishes and completes all orders for product 1, whereas, in region 2, focal manufacturer 2 prioritizes orders for product 1 and so on.

**Table 5.** Priority from scenario 1.

| Region Priority | FM 1 | FM 2 | FM 3 |
|---|---|---|---|
| Product 1—biomodel | High | Medium | Medium |
| Product 2—cutting guide | Medium | High | Low |
| Product 3—implant | Low | Low | High |

Source: own elaboration.

From the initial conditions shown in the scenario, the input data were recorded, where the orders are generated at the beginning of each week, as shown in Table 6.

**Table 6.** Monthly orders record for the traditional scenario.

| | FM1/Product1 | | | FM 2/Product 2 | | | FM 3/Product 3 | | |
|---|---|---|---|---|---|---|---|---|---|
| Hour | R1 | R2 | R3 | R1 | R2 | R3 | R1 | R2 | R3 |
| 7 | 12 | 5 | 2 | 5 | 2 | 1 | 2 | 1 | 1 |
| 175 | 11 | 6 | 1 | 5 | 2 | 1 | 2 | 1 | |
| 343 | 11 | 5 | 2 | 5 | 2 | 1 | 2 | 1 | |
| 511 | 11 | 6 | 2 | 4 | 3 | | 2 | 1 | |
| | 45 | 22 | 7 | 19 | 9 | 3 | 8 | 4 | 1 |

Source: own elaboration.

4.1.2. Scenario 2—Centralized Additive Manufacturing Supply Chain

The second scenario corresponds to the supply chain with centralized additive manufacturing, where the existing groups of suppliers assume the additive process, i.e., there is a change in the production approach (subtractive to additive); however, communication and transportation with the other links remain structurally the same [16]. The main changes would be in the internal processes and chain management, which from its configuration would disaggregate a previous link to the suppliers of drugs, biomedical equipment, medical devices, and support services, including the supply of raw materials for their manufacture in cases where AM is applicable [22].

Figure 9 shows a chain made up of a supplier, a focal manufacturer, and a distributor as in the base model. The difference lies in the three regions centralized in region 1, where the three products are produced. The policy is to serve the closest region, region 1, then region 2, and finally, region 3, which is the furthest away.

Once again, in order to simulate the scenario, it was necessary to modify the base model. The demand is represented in Figure 10, which shows the region demand request for the three products, which ultimately constitutes the total demand for the region, and so on with the other regions. It also shows the policies of priority of attention for product and region and the link's parameters in the supply chain.

The supply chain structure remains similar to the base model, as shown in Figure 8. However, the necessary vectors for the model operation are increased in this scenario. Vectors one, two, and three correspond to the attention of region 1 with the priority of products one, two, and three, creating the variable of the order of production of region 1 (Order_ProdR1). Vectors four, five, and six correspond to the attention of region 2 and the same product priority creating the variable of the production order of region 2 (Order_ProdR2). Finally, vectors seven, eight, and nine correspond to the attention of region 3, keeping the same product priority and generating the variable of production order of region 3 (Order_ProdR3). All the production orders were integrated into (Order_Prod), containing the nine vectors.

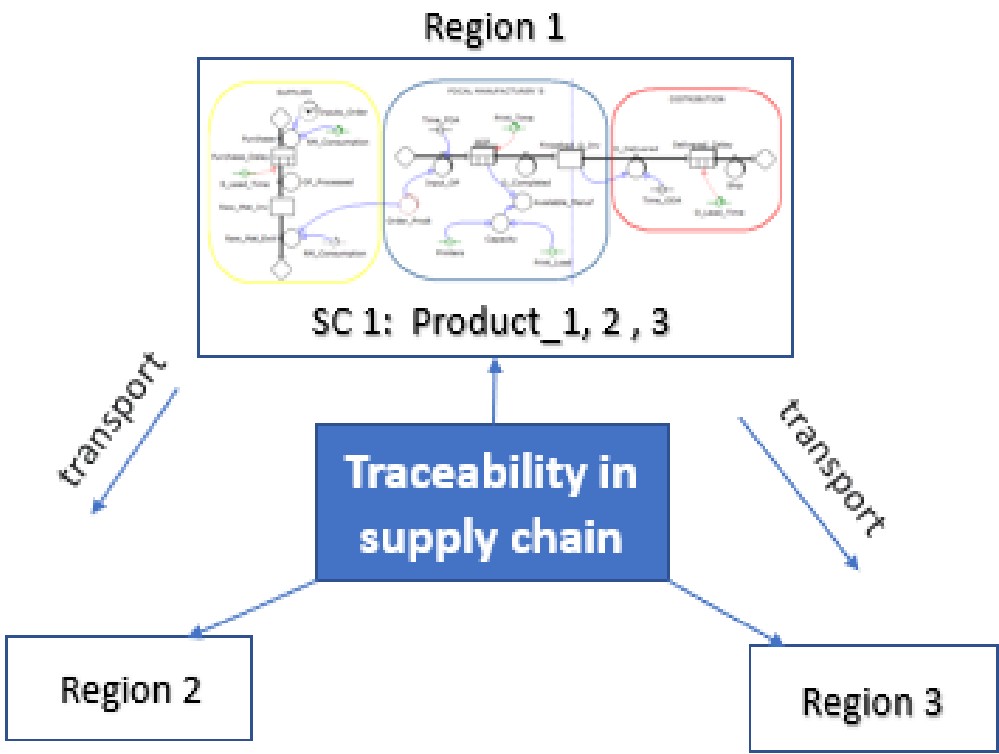

**Figure 9.** Scenario 2 representation—a centralized additive supply chain. Source: own elaboration.

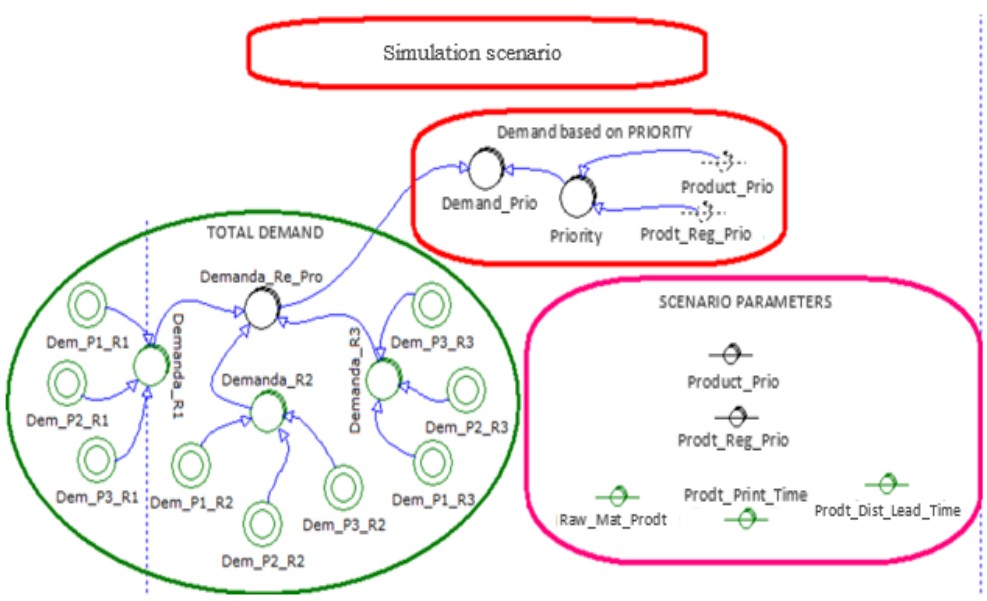

**Figure 10.** Total demand—scenario 2. Source: own elaboration by Evolution software.

Two key aspects are highlighted in this modeling. The first is that each region's distribution time (Prodt_D_Lead_Time) is different. The second corresponds to the printing time (Print_Time), which should establish the vector's position according to the product type being processed and the attention policy established. Figure 11 presents the policy of the regions.

In order to contrast the models and determine the AM's impact in terms of chain cycle time, the annual and monthly demand distribution data of the scenario were kept in the simulation of scenario 2 (Tables 6 and 7).

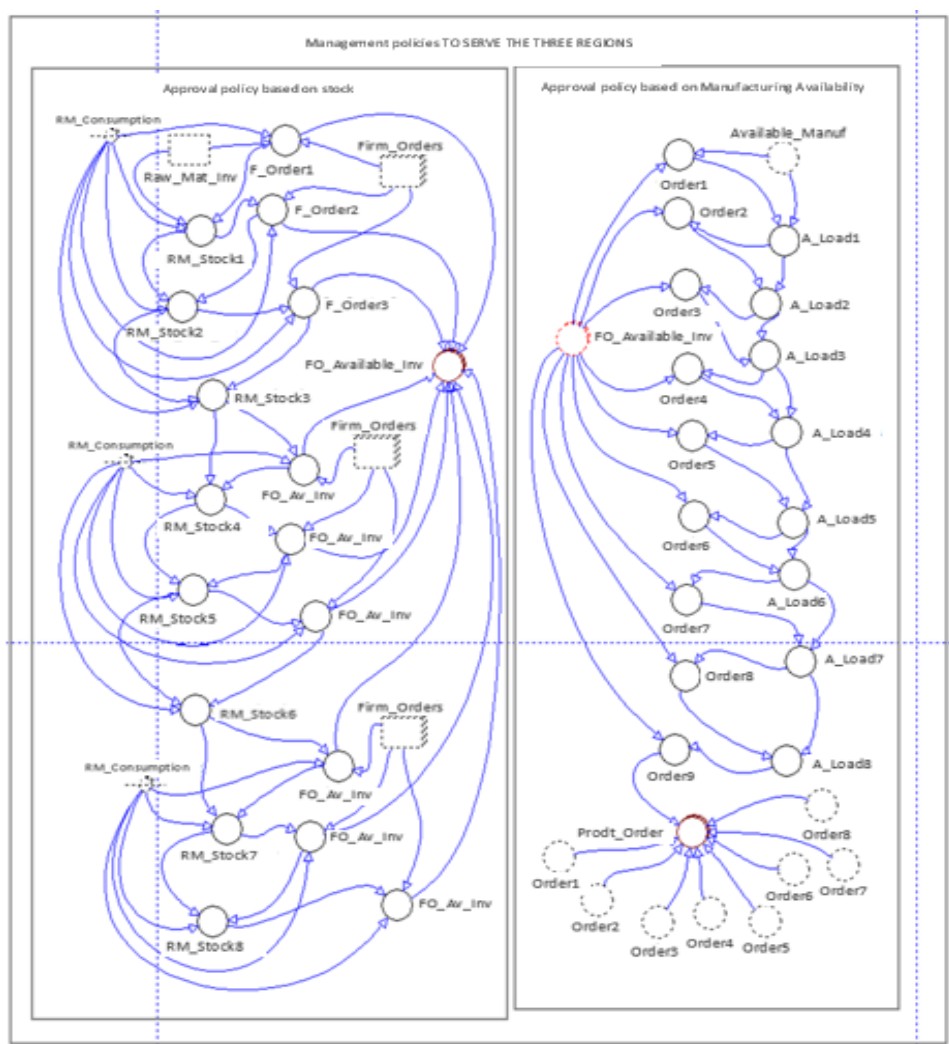

**Figure 11.** Management policies to serve the regions—scenario 2. Source: own elaboration. Evolution software.

**Table 7.** Scenario 2 conditions.

|  | FM 1 |
| --- | --- |
| Material consumption | Grams |
| Product 1 | 81 |
| Product 2 | 14 |
| Product 3 | 21 |
| Production conditions | Units |
| Number of machines/printers | 3 |
| Total load per machine | 1 |
| Chain times | Hours |
| Procurement times |  |
| Raw material | 120 |
| Printing times |  |
| Product 1 | 12 |
| Product 2 | 4 |
| Product 3 | 5 |
| Distribution lead time |  |
| Region 1 | 24 |
| Region 2 | 48 |
| Region 3 | 72 |

Source: own elaboration.

Table 7 shows the capacity, material, and time conditions of scenario 2. In this case, the manufacturer is located in region one and is responsible for producing the three products for distribution in the other areas. Procurement times are 120 h, and printing times are associated with the size of the product whereas distribution times are due to proximity. For example, when delivery occurs in the same region, the estimated time is 24 h, for the next closest area it is 48 h, and for the farthest region, it would take 72 h to deliver the final product.

The priority policy is detailed in Table 8. First, it is necessary to meet the demand for the three products in region 1, then region 2, and finally, region three.

**Table 8.** Scenario 2 conditions.

|  | Region 1 | Region 2 | Region 3 |
|---|---|---|---|
| Product 1—biomodel | High | Medium | Low |
| Product 2—cutting guide | High | Medium | Low |
| Product 3—implant | High | Medium | Low |

Source: own elaboration.

From the initial conditions shown in the scenario, the input data were recorded, where the orders are generated at the beginning of each week, as shown in Table 9.

**Table 9.** Monthly order entry of the SC scenario with centralized AM.

| | Region 1 | | | Region 2 | | | Region 3 | | |
|---|---|---|---|---|---|---|---|---|---|
| Product | P1 | P2 | P3 | P1 | P2 | P3 | P1 | P2 | P3 |
| Priority | High | High | High | Medium | Medium | Medium | Low | Low | Low |
| Hour | | | | | Number of Products | | | | |
| 7 | 12 | 6 | 2 | 6 | 3 | 1 | 2 | 1 | 1 |
| 175 | 11 | 5 | 2 | 6 | 3 | 1 | 2 | 1 | 0 |
| 343 | 11 | 4 | 2 | 5 | 2 | 1 | 2 | 1 | 0 |
| 511 | 11 | 4 | 2 | 5 | 1 | 1 | 1 | 0 | 0 |
| | 45 | 19 | 8 | 22 | 9 | 4 | 7 | 3 | 1 |

Source: own elaboration.

4.1.3. Scenario 3—Decentralized Additive Supply Chain

The third scenario includes additive manufacturing in a decentralized supply chain [23,24]. In this case, the additive process is assumed by the health service provider institutions (IPSs); i.e., there would be production spaces (data collection, printing, and finishing) of medical devices or elements, which implies an effort in the reconfiguration of internal processes, chain management, and existing links. At the initial stage, the traditional groups of suppliers would be retained since they are required to respond to medical emergencies and less complex procedures where it is not appropriate to use AM. However, these suppliers may or may not assume the role of supplying inputs for the printing process within the IPSs, which means that these institutions require new services in materials, digitization, and maintenance of 3D printing equipment. When the IPSs incorporate these services, they stop participating in the chain only as service providers and become involved in manufacturing. Figure 12 shows the printing process in each region where there is demand, eliminating any external transportation and physical storage.

With this approach, three chains are proposed, each with a supplier, a manufacturer, and an independent demand, which are linked to a logistics operator that distributes the demand according to the needs of each region (in this case, three regions) and also keeps control of the firm orders, those for production, those in production, and those set to be delivered. These previous orders are also stored in an order delivery record to monitor the performance and operation of each manufacturer.

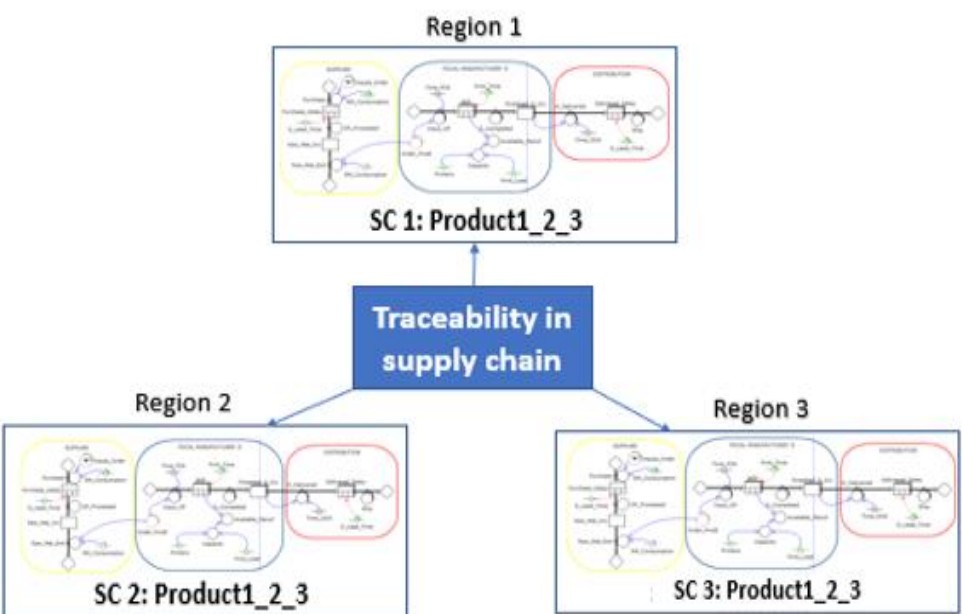

**Figure 12.** Decentralized additive supply chain. Source: own elaboration.

These considerations imply changes in the base model, particularly in the vectorization and establishment of a logistics operator. Figure 7 represents the modeling of the logistics operator section added to the original model. It represents the interaction of the demands and the control exercised over the productions to store them in the fulfilment history.

The previous modification made it necessary to define the demand managed by the logistics operator, as shown in Figure 13. In this case, the product priority is also maintained, i.e., product 1 is served first, then product 2, and finally, product 3, in each of the focal manufacturers/regions.

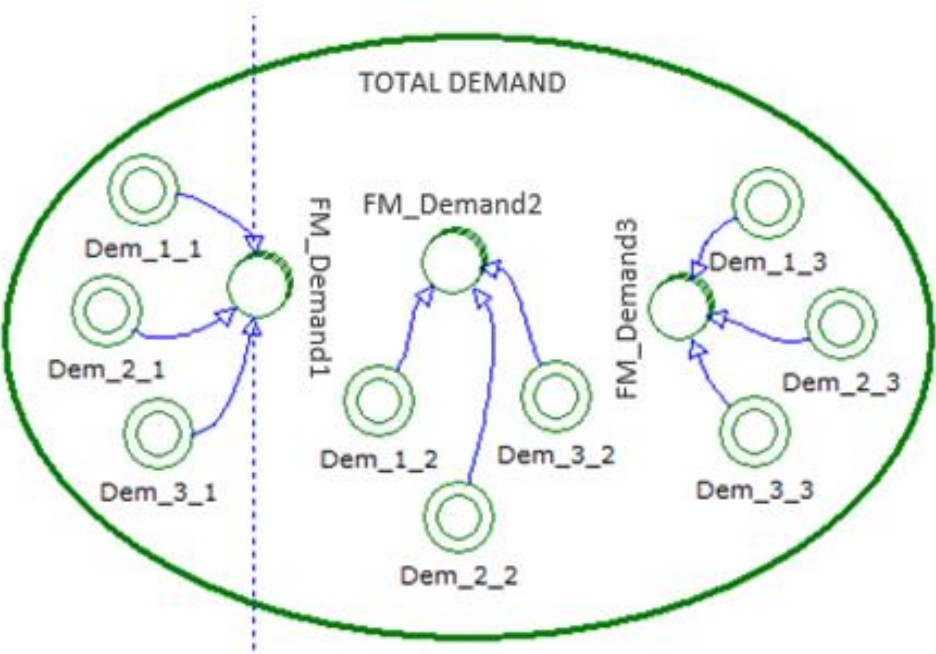

**Figure 13.** Demand managed by the logistics operator. Source: own elaboration. Evolution software.

With the demand representation, it is possible to propose producing new products that are served globally (three for each region) with different raw material consumption, printing times, manufacturer capacities, and distribution times. Likewise, the regions

where the machines/printers operate are determined, as shown in Figure 14, where each is associated with a specific region.

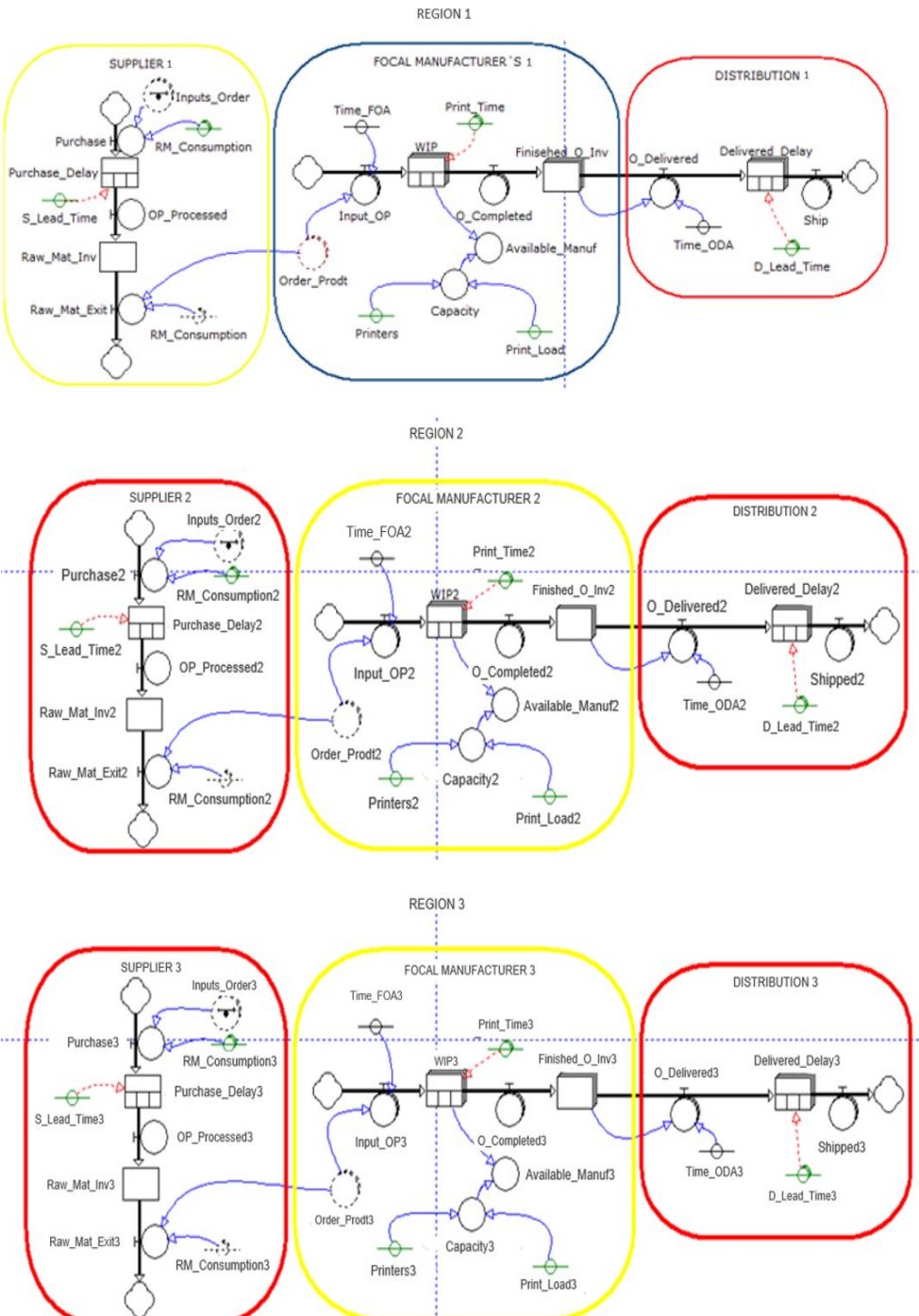

**Figure 14.** Structure of scenario 3 by region. Source: own elaboration. Evolution software.

In addition, for the structural operation of the focal manufacturer, a policy was defined that operates independently for each region, with the possibility of each region taking action differently. Figure 15 shows this policy, which depends on stock and capacity acceptance. In the case of acceptance by stock, the quantities of available and existing raw materials are checked against the orders to meet and the products to produce. As for acceptance by capacity, the production order variables and the priority criteria are reviewed for each case.

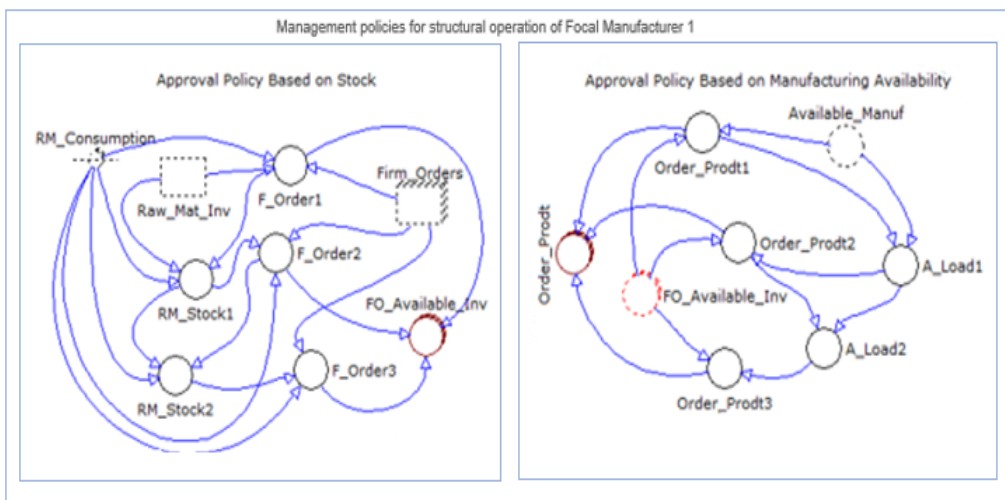

**Figure 15.** The structural operating policy of focal manufacturer 1. Source: own elaboration. Evolution software.

This regionalized scenario allows for greater flexibility in achieving alternatives and satisfying demand, given that each manufacturer operates independently according to its operating parameters and structural policy.

Regionalization is also observed in raw materials, which meet the same conditions as the base model, i.e., a single regional supplier that manages the manufacturer's inventory and a distribution company that delivers the product as it is produced.

This scenario presents the model amplification, which allows visualizing a greater functionality of the logistics operator since it controls the different business units framed in simplified supply chains of a single supplier, manufacturer, and distribution network.

Having the same objective from the first two scenarios made it necessary to retain the annual and monthly demand distribution data (Tables 6 and 7) to simulate the discussed scenario.

Table 10 shows the capacity, material, and timing conditions of scenario 3. There are three manufacturers and three supply chains, each located in a region, in charge of producing the three products and supplying the same territory. The supply times are 120 h in each chain, the printing times are associated with the size of the product, and the distribution times correspond, in this case, to 24 h, since in each region there is a chain that supplies the three products throughout the territory.

**Table 10.** Monthly order entry of the SC scenario with centralized AM.

|  | FM 1 | FM 2 | FM 3 |
|---|---|---|---|
| Material consumption | Grams | Grams | Grams |
| Product 1 | 81 | 81 | 81 |
| Product 2 | 14 | 14 | 14 |
| Product 3 | 21 | 21 | 21 |
| Production conditions | Units | Units | Units |
| Number of machines/printers | 1 | 1 | 1 |
| Total load per machine | 1 | 1 | 1 |
| Chain times | Hours | Hours | Hours |
| Procurement times |  |  |  |
| Raw material | 120 | 120 | 120 |
| Printing times |  |  |  |
| Product 1 | 12 | 12 | 12 |
| Product 2 | 4 | 4 | 4 |
| Product 3 | 5 | 5 | 5 |

**Table 10.** *Cont.*

|  | FM 1 | FM 2 | FM 3 |
|---|---|---|---|
| Distribution lead time |  |  |  |
| Region 1 | 24 |  |  |
| Region 2 |  | 24 |  |
| Region 3 |  |  | 24 |

Source: own elaboration.

For scenario 3, the priority policy is based on the product type, given that all three are produced in each region. For example, in region 1, product 1 is made first, then product 2, and finally, product 3.

The input data were recorded based on the initial conditions related to the scenario. The corresponding orders are generated at the beginning of each week, as shown in Table 11, changing the product priority for each region concerning scenario 2.

**Table 11.** Monthly order record for scenario 3.

|  |  |  | Region 1 | | | Region 2 | | | Region 3 | | |
|---|---|---|---|---|---|---|---|---|---|---|---|
|  | Product | | P1 | P2 | P3 | P1 | P2 | P3 | P1 | P2 | P3 |
|  | Priority | | High | Medium | Low | High | Medium | Low | High | Medium | Low |
| Week | Day | Hour | | | | Number of Products | | | | | |
| 1 | 1 | 7 | 12 | 6 | 2 | 6 | 3 | 1 | 2 | 1 | 1 |
| 2 | 8 | 175 | 11 | 5 | 2 | 6 | 3 | 1 | 2 | 1 | 0 |
| 3 | 15 | 343 | 11 | 4 | 2 | 5 | 2 | 1 | 2 | 1 | 0 |
| 4 | 22 | 511 | 11 | 4 | 2 | 5 | 1 | 1 | 1 | 0 | 0 |
|  |  |  | 45 | 19 | 8 | 22 | 9 | 4 | 7 | 3 | 1 |

Source: own elaboration.

*4.2. Scenario Simulation*

Based on the initial conditions of each model and scenario, the models were run in the Evolution software to obtain the results that are organized as follows:

- Lead-time analysis indicates the time elapsed from the reception of the first order in week one until all orders of the product category are recorded in the order record.
- Analysis of available capacity and orders in production and inventory.
- Analysis of the behavior of raw material and finished product inventories.

4.2.1. Chain Lead-Time Analysis

The first result is associated with the fulfilment of the projected monthly demand for each product in each region, i.e., it was determined how long it would take to complete all the orders and record them in the order record. The lead time was set at the time the entire demand was fulfilled. For example, the demand for product 1 in region 1 in any scenario corresponds to 45 units recorded in hour 1. The demand starts its journey through the supply chain and is fully distributed and delivered at hour 748 in scenario 1, hour 671 in scenario 2, and hour 764 in scenario 3.

Table 12 was constructed from the results, which summarizes the supply chain lead time for each scenario, grouping the products as a total in each region. In addition, the variation of weeks between scenarios 1 and 2, and 1 and 3 was calculated to contrast the TSC approach with the ASC. Additionally, the divergence between scenarios 2 and 3 was compared to understand the changes between the centralized and decentralized supply chains.

**Table 12.** Lead time and scenario variation.

| Region | Region 1 | | | Region 2 | | | Region 3 | | |
|---|---|---|---|---|---|---|---|---|---|
| Product | **P1R1** | **P2R1** | **P3R1** | **P1R2** | **P2R2** | **P3R2** | **P1R3** | **P2R3** | **P3R3** |
| Total Units | 45 | 19 | 8 | 22 | 9 | 4 | 7 | 3 | 1 |
| Lead-time FM1 (hours) | 748 | 716 | 693 | 1048 | 672 | 723 | 1163 | 577 | 159 |
| Lead-time FM2 (hours) | 671 | 588 | 674 | 724 | 736 | 742 | 803 | 802 | 804 |
| Lead-time FM3 (hours) | 763 | 833 | 881 | 728 | 559 | 733 | 670 | 520 | 189 |
| Variation FM2—FM1 (%) | −11.48 | −21.77 | −2.82 | −44.75 | 8.7 | 2.56 | −44.83 | 28.05 | 80.22 |
| Variation FM3—FM1 (%) | −2.01 | −16.34 | −27.13 | 30.53 | 16.82 | −1.38 | 42.39 | 9.88 | −18.87 |
| Variation FM2—FM3 (%) | 12.06 | 29.41 | 23.5 | 0.55 | −31.66 | −1.23 | −19.85 | −54.23 | −325.4 |

Source: own elaboration.

The contrasting scenarios show how the structural changes in the chain and the manufacturers' location reduce the total product delivery times for the cases depending on the region. For example, procurement, production, and distribution times of the 45 units of product 1 for region 1 are reduced by 11.48% in scenario 2 compared to scenario 1, whereas scenario 3 compared to scenario 1 decreased the hours by 2%. Finally, comparing scenarios 2 and 3, the lead time increased by 12.06%.

After analyzing the variation from scenarios 2 and 3, which contrasted a centralized AM supply chain with a TSC, it was possible to identify five negative variations out of nine (P1R1, P2R1, P3R1, P1R2, P1R3). The lead time was reduced mainly in region 1. On the other hand, times increased in the other four cases, two in the second region (P2R2, P3R2) and the remaining in the third region (P2R3, P3R3).

After analyzing the variation from scenarios 3 and 1, which contrasted a decentralized AM supply chain with a TSC, it was possible to identify five negative variations out of nine (P1R1, P2R1, P3R1, P3R2, P3R3). The analysis indicates that the lead time is then again reduced, and there is an increase in terms of time for the remaining four cases (P1R2, P2R2, P1R2, P2R2).

Finally, as for the variation from scenarios 2 and 3, which contrasted a centralized AM supply chain with a decentralized AM supply chain, researchers observed a reduction in times for products and regions related to the following cases: P2R2, P3R2, P1R3, P2R3, and P3R3. At the same time, it was possible to identify a decrease for the following cases: P1R1, P2R1, P3R1, P3R1, and P1R2.

In all cases, the variations are mainly associated with the initial conditions described above; the individual procurement, production, and distribution times, as well as the priority policies, impact the decision-making process regarding production planning. It should be noted that the chain's installed capacity remained the same in the three scenarios (3 machines with a capacity of 1 unit per machine). Procurement times correspond to 120 h, printing times are maintained depending on the product's weight, and distribution times vary according to proximity, with deliveries ranging from 24 to 72 h. In order to understand the variations in a better way, researchers analyzed the following variables: available capacity, orders in production and pending delivery, and inventory levels. The latter is better detailed in the following paragraphs.

### 4.2.2. Behavioral Analysis of Orders, Inventories, and Available Capacity

Figure 16 contrasts the behavior of production orders for each scenario. In this way, researchers observed the operation of the priority policy established for each case. Scenario 1 shows the three manufacturers producing the product they are in charge of for shipment to the destination region. Region 1, being in charge of product 1, concentrates most of the orders, first handling orders from its own region and continuing with regions 2 and 3; it finishes the orders in production in hour 1100. In scenario 2, the last production orders occur in hour 730, where all three types of products are handled simultaneously since only one manufacturer is located and is in charge of manufacturing all products. In scenario 3, a similar behavior to the first scenario is observed. However, in this case, the last production

order occurs at hour 900, 200 h earlier than in scenario 1 since the manufacturers are in charge of producing all three products and not just one.

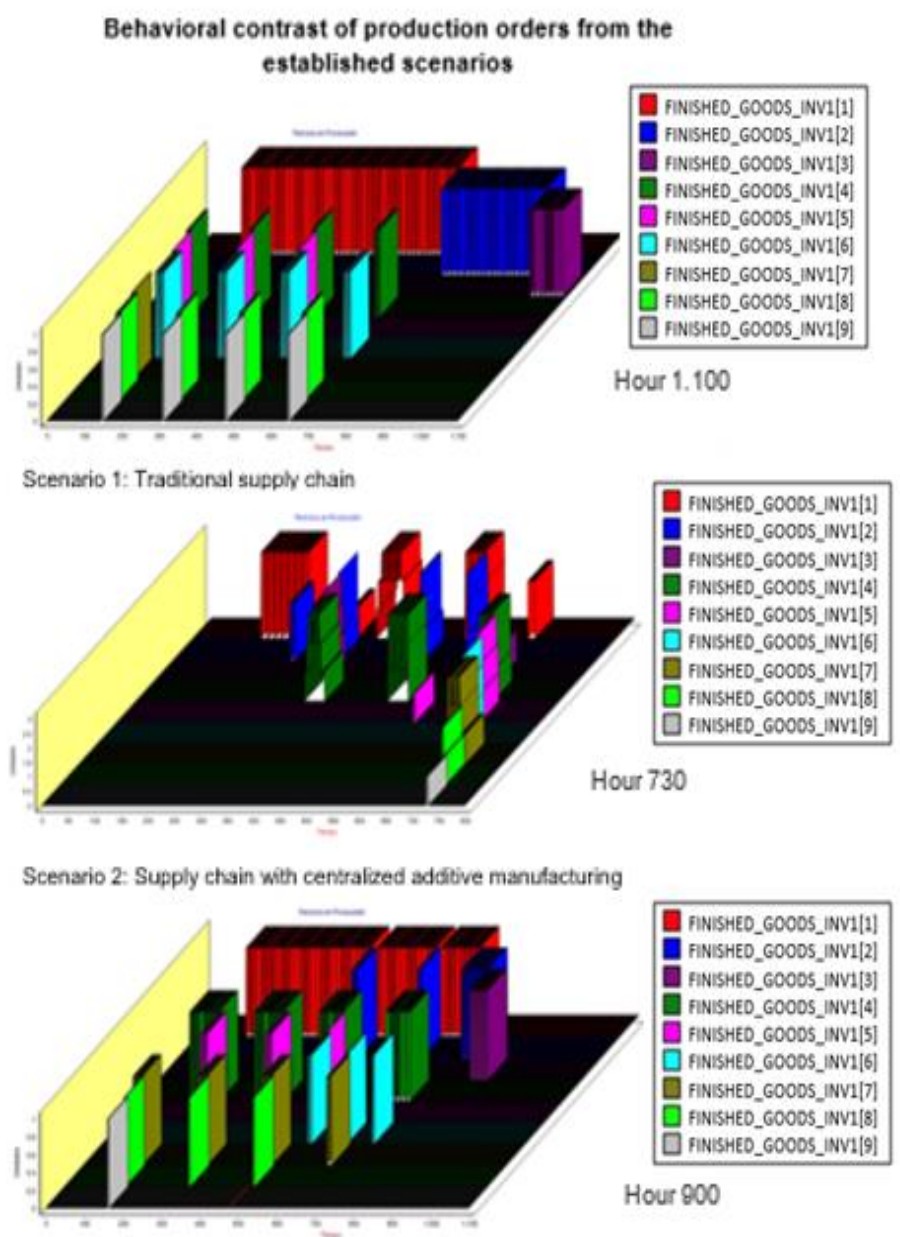

**Figure 16.** Behavioral contrast of production orders in the established scenarios. Source: own elaboration. Simulation results obtained from Evolution software.

Researchers analyzed the graphs summarized in Figure 17 to determine the behavior of the available capacity. As for scenarios 1 and 3, where there are three manufacturers, there is a clear distinction between the product types and demand. For example, in the first scenario, where the manufacturer in region 1 (FM1R1) is responsible for the production and distribution of product 1 to all regions, and hence, is responsible for 62.71% of the demand, whereas the manufacturer in region 2 (FM2R2) has 26.27% and the manufacturer in the third region (FM3R3) has 11.02%, which means that FM2R2 and FM3R3 have a greater available capacity. In the third scenario, even though each manufacturer is capable of producing and delivering within its region, the percentages of demand served by each

region are very similar to those of the products (61.02%, 29.66%, and 9.32%), which means that in regions 2 and 3 there is a greater available capacity to meet orders.

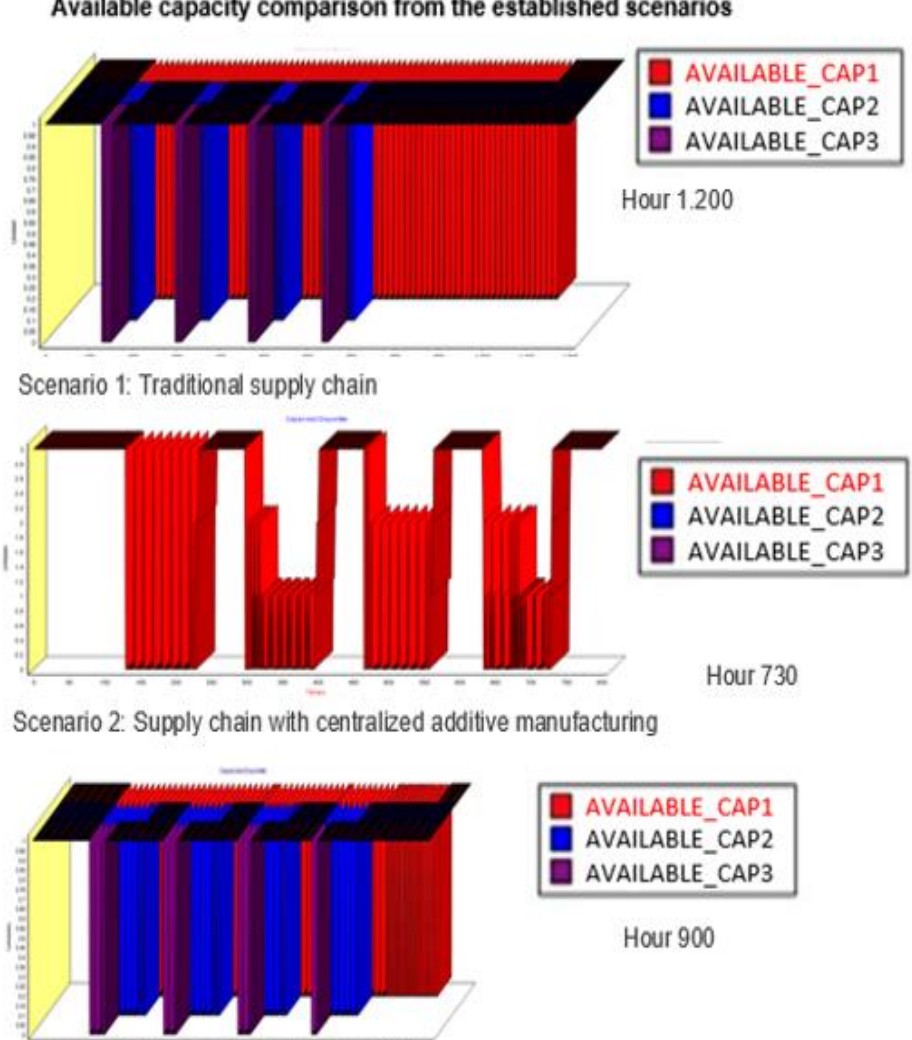

**Figure 17.** Available capacity comparison from the established scenarios. Source: own elaboration. Simulation results obtained from Evolution software.

Regarding times, it is possible to evidence that FM1 becomes available in hour 731 in the second scenario, in hour 900 in the third scenario, and the same happens in hour 1.200 during the first scenario. The latter shows that the additive supply chain scenarios have a greater capacity to meet higher demand.

Likewise, the pending delivery orders were reviewed in the scenarios presented in Figure 18. In the first scenario, the pending delivery orders go up to approximately 1150 h, and are ready to be distributed to the regions according to the demand. The maximum number of orders accumulated for delivery is six. Scenario 2, on the other hand, presents more significant fluctuations with periods at zero since it meets the demand before registering the start of a new week. However, it reaches peaks of 12 products on backorder, i.e., 6 more than the first scenario and 7 more than the third one.

In scenario 1, some products have off-peak periods, but in others, such as P1R1, those products are permanently back-ordered until demand is met after 700 h.

In scenario 3, for every hour there are, on average, two orders pending delivery, which remains constant in almost all hours, except for products P2R1 and P3R1, due to the demand each represents. In this case, the last backorder is at hour 870.

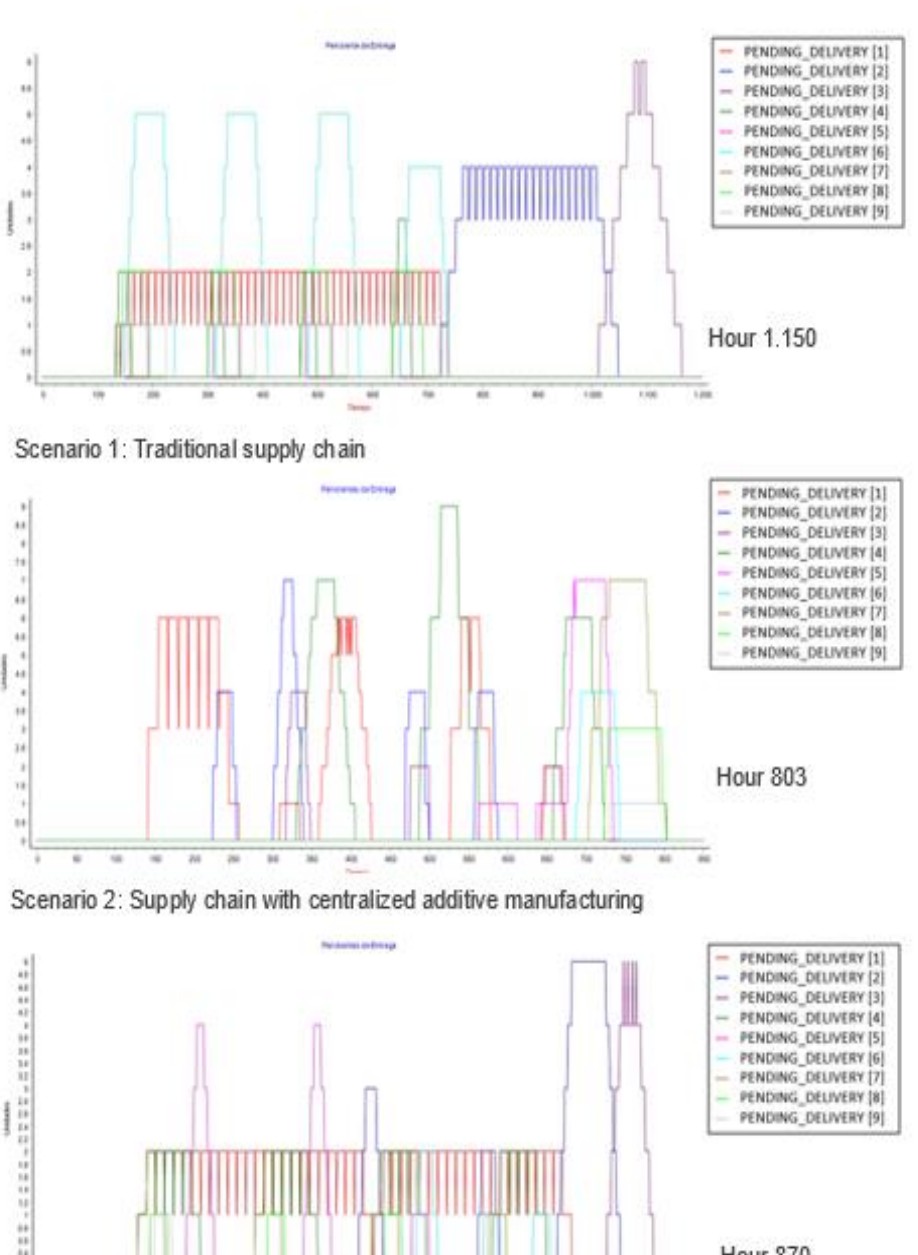

**Figure 18.** Behavioral contrast on pending delivery orders from the established scenarios. Source: own elaboration. Simulation results obtained from Evolution software.

The scenario with the lowest backlog is scenario 3, whereas scenario 2 is the scenario where orders are dispatched for delivery the fastest. Both scenarios include additive manufacturing in their production processes, which generates flexibility of response to meet the demand and distribute to each region as suitable.

In order to verify the behavior of the pending delivery orders, the behavior of the finished product inventory was projected, as shown in Figure 19.

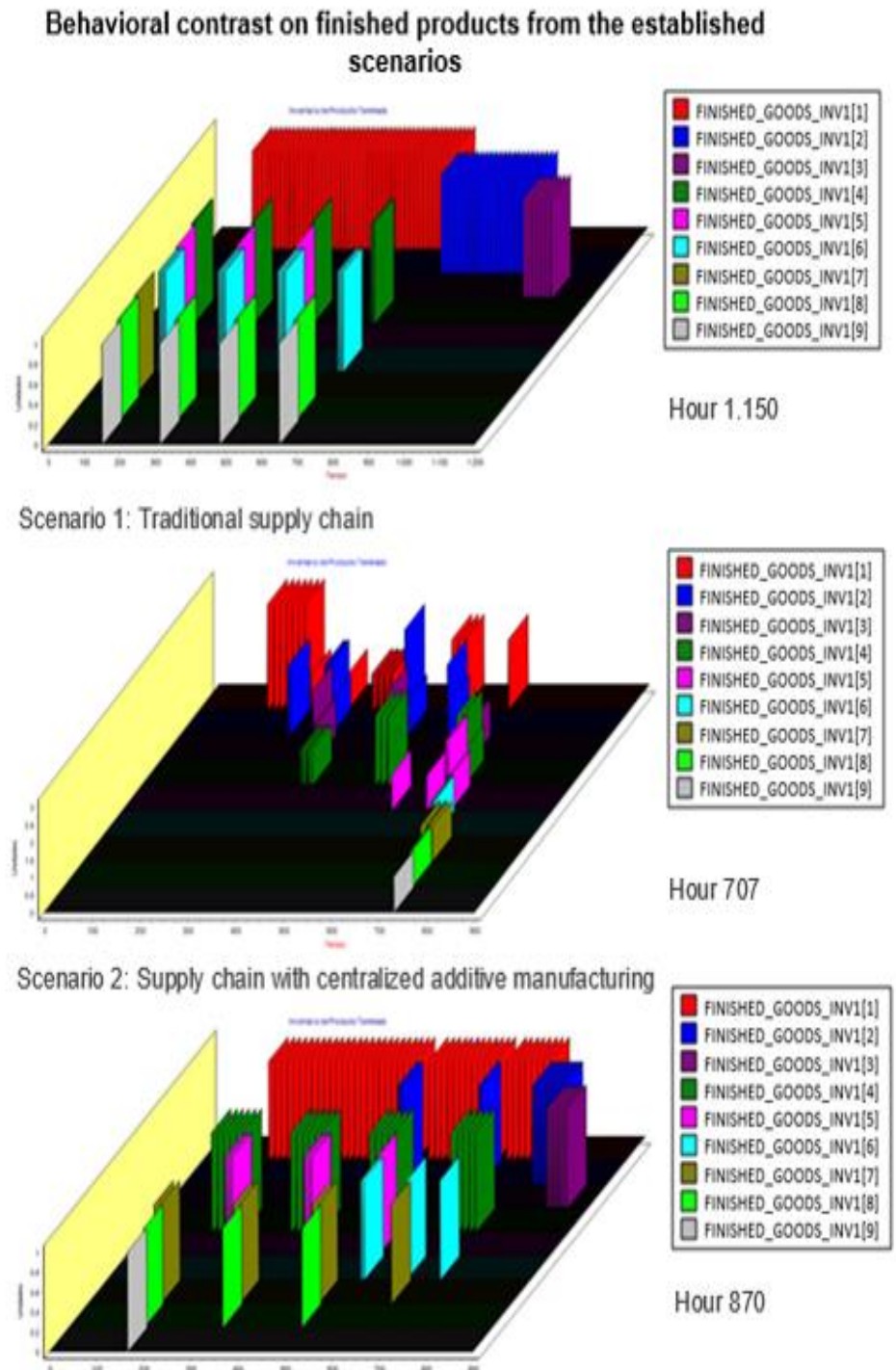

**Figure 19.** Behavioral contrast on finished product inventory from the established scenarios. Source: own elaboration. Simulation results obtained from Evolution software.

The raw material inventory was also simulated, as shown in Figure 20. It is possible to observe that, even though demand remains equal in all three cases, the behavior of orders in production and the available capacity of each manufacturer generates the following variations:

— In scenario 1, when there are new orders generated, FM1R1 starts requiring 1600 units of raw material, then it accumulates orders and reaches a peak of 2900 units to complete production. In comparison, FM2R2 and FM3R3 have an inventory of less than 100 units that runs out in a few hours since their demand is much lower than that of product 1.

— In scenario 2, the behavior is divided for each week, where average peaks of 1800 units are reached and then decrease, reaching off-peaks at zero when order production has finished and new orders have not been generated. One of the reasons for the previous situation is that only one manufacturer manages a single inventory of raw materials.

— In scenario 3, the material's behavior reaches a maximum of 1100 units in the case of FM1R1 because it accumulates the most significant quantity of products and demand, whereas in FM2R2 and FM3R3, the amounts reach 650 and 200, respectively.

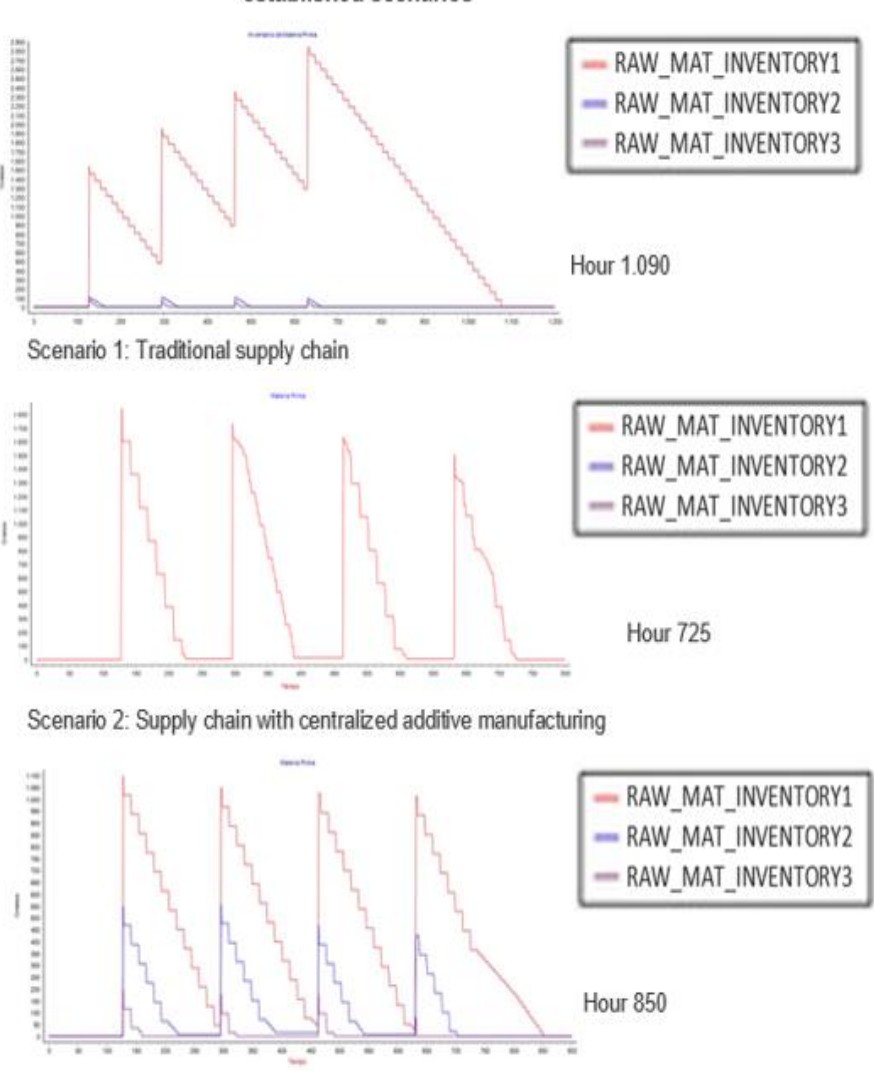

**Figure 20.** Behavioral contrast of raw material inventory from the established scenarios. Source: own elaboration. Simulation results obtained from Evolution software.

The hours in which the raw material inventory is determined varies significantly from scenario 1 to scenarios 2 and 3. Although manufacturers require 1090 h to finish all units in the first scenario, in the second and third scenarios, 725 and 850 are needed, respectively.

The latter demonstrates how the additive supply chain scenarios represent a better balance in inventory accumulation.

## 5. Discussion

A case study was defined to explain the impact of implementing additive manufacturing through supply chain configurations. Hence, three scenarios were proposed:

1. Traditional supply chain;
2. Supply chain with centralized additive manufacturing;
3. Supply chain with decentralized additive manufacturing.

Based on data from the Colombian context and the literature on the usage of medical devices (biomodels, cutting guides, and implants), the lead times, available capacities, and inventories of raw materials and finished products were obtained.

Researchers designed a modeling baseline from the results obtained from the simulations to continue with the variation of parameters and values based on the particular contexts of each industry. The previous results allowed researchers to understand the benefits for companies and, even more so, for supply chains of adopting this approach in the transformation process. This starting point can vary significantly in terms of the actors in the chain since scenarios could be examined in which another link could take on the additive process, which, in short, would mean simplifying the number of links that make up the chain. In addition, we could begin to explore scenarios in which distribution takes on other types of responsibilities and modes of transportation. The supplier, for example, could carry out part of the additive process, and even the client itself could be in charge of production. However, these are alternatives that will be explored in future research.

By evaluating the available capacity, orders in production, and inventories, it was possible to define the maximum quantities of raw material units required in each scenario. It was shown that scenario 1 accumulated orders from previous weeks due to its low available capacity, which made it require up to 100% more units in the third week than the other scenarios. As for scenario 2, the conditions explained in the case study are considered the most appropriate in proportion to the benefits it showed in the short term. However, future conditions and the trend towards creating printing farms and atomized chains should be considered, as they will allow for greater customization and rapid changes in production when there are low-volume orders, which is similar to the third scenario.

## 6. Conclusions

Based on all the product iterations carried out throughout the simulated months, it is possible to notice that, although the production time is longer in additive manufacturing compared to traditional manufacturing, the cycle time and total demand closure were lower than in traditional manufacturing. In addition, it was observed that the AM performance is significantly better in conditions of lower demand, which can be attributed to the characteristics of customization and small batches that this type of production approach implies.

The total demand is distributed in three regions with the three products. When the chain is managed in a traditional way, where each manufacturer produces a specific product, the results showed that lead times are concentrated in the first company from the first region since product 1 corresponds to 60% of the demand, which is the region that requires the most significant quantity.

As for the chain where only one manufacturer is in charge of producing and distributing the total demand with additive manufacturing techniques, which is located in the region with the highest demand, the response time is reduced, increasing the company's available capacity in six of the nine cases analyzed.

Finally, for a decentralized chain, focal manufacturers are without any exchange between regions. The latter generates a better, more accurate response in each one, and the bottleneck continues to be in the first region. The latter also reduces response times and balances the operations between them.

Finally, it should be noted that the optimal scenario for including the AM in the SSC depends on the volume of demand and the cost-effectiveness assessment it can offer to the actor that assumes it.

**Author Contributions:** Conceptualization, J.N.R. and S.M.V.-A.; methodology, J.N.R. and H.H.A.S.; software, J.N.R. and H.H.A.S.; validation, J.N.R. and H.H.A.S.; formal analysis, J.N.R. and H.H.A.S.; investigation J.N.R. and A.O.; resources, J.N.R. and A.O.; data curation, J.N.R. and H.H.A.S.; writing—review and editing, J.N.R., H.H.A.S., A.O. and S.M.V.-A.; visualization J.N.R.; supervision A.O. and H.H.A.S.; project administration, J.N.R. and A.O. All authors have read and agreed to the published version of the manuscript.

**Funding:** This research received no external funding.

**Data Availability Statement:** All data are available upon request.

**Conflicts of Interest:** The authors declare no conflict of interest.

## Appendix A

Abe, F., Costa Santos, E., Kitamura, Y., Osakada, K., & Shiomi, M. (2003). Influence of forming conditions on the titanium model in rapid prototyping with the selective laser melting process. Proceedings of the Institution of Mechanical Engineers, Part C: Journal of Mechanical Engineering Science, 217(1), 119–126. https://doi.org/10.1243/095440603762554668

Ahn, D.-G., Lee, J.-Y., & Yang, D.-Y. (2006). Rapid Prototyping and Reverse Engineering Application for Orthopedic Surgery Planning. Journal of Mechanical Science and Technology, 20(1), 19–28. https://doi.org/10.1007/BF02916196

Al-Ahmari, A., Nasr, E. A., Moiduddin, K., Alkindi, M., & Kamrani, A. (2015). Patient specific mandibular implant for maxillofacial surgery using additive manufacturing. 2015 International Conference on Industrial Engineering and Operations Management (IEOM), 1–7. https://doi.org/10.1109/IEOM.2015.7093788

Arcaute, K., & Wicker, R. B. (2008). Patient-Specific Compliant Vessel Manufacturing Using Dip-Spin Coating of Rapid Prototyped Molds. Journal of Manufacturing Science and Engineering, 130(5), 051008. https://doi.org/10.1115/1.2898839

Attar, H., Calin, M., Zhang, L. C., Scudino, S., & Eckert, J. (2014). Manufacture by selective laser melting and mechanical behavior of commercially pure titanium. Materials Science and Engineering: A, 593, 170–177. https://doi.org/10.1016/j.msea.2013.11.038

Bauermeister, A. J., Zuriarrain, A., & Newman, M. I. (2016). Three-Dimensional Printing in Plastic and Reconstructive Surgery. Annals of Plastic Surgery, 77(5), 569–576. https://doi.org/10.1097/SAP.0000000000000671

Beaucamp, A. T., Namba, Y., Charlton, P., Jain, S., & Graziano, A. A. (2015). Finishing of additively manufactured titanium alloy by shape adaptive grinding (SAG). Surface Topography: Metrology and Properties, 3(2), 024001. https://doi.org/10.1088/2051-672X/3/2/024001

Berretta, S., Evans, K., & Ghita, O. (2018). Additive manufacture of PEEK cranial implants: Manufacturing considerations versus accuracy and mechanical performance. Materials & Design, 139, 141–152. https://doi.org/10.1016/j.matdes.2017.10.078

Berry, E., Brown, J. M., Connell, M., Craven, C. M., Efford, N. D., Radjenovic, A., & Smith, M. A. (1997). Preliminary experience with medical applications of rapid prototyping by selective laser sintering. Medical Engineering & Physics, 19(1), 90–96. https://doi.org/10.1016/S1350-4533(96)00039-2

Bibb, R., Eggbeer, D., Evans, P., Bocca, A., & Sugar, A. (2009). Rapid manufacture of custom-fitting surgical guides. Rapid Prototyping Journal, 15(5), 346–354. https://doi.org/10.1108/13552540910993879

Bill, J. S., Reuther, J. F., Dittmann, W., Kübler, N., Meier, J. L., Pistner, H., & Wittenberg, G. (1995). Stereolithography in oral and maxillofacial operation planning. International Journal of Oral and Maxillofacial Surgery, 24(1), 98–103. https://doi.org/10.1016/S0901-5027(05)80869-0

Blaya, F., Pedro, P. S., Lopez-Silva, Julia., D'Amato, Roberto., Juanes, J. A., & Lagándara, J. G. (2017). Study, design and prototyping of arm splint with additive manufacturing process. Proceedings of the 5th International Conference on Technological Ecosystems for Enhancing Multiculturality—TEEM 2017, 1–7. https://doi.org/10.1145/3144826.3145407

Bose, S., Ke, D., Sahasrabudhe, H., & Bandyopadhyay, A. (2018). Additive manufacturing of biomaterials. Progress in Materials Science, 93, 45–111. https://doi.org/10.1016/j.pmatsci.2017.08.003

Brito, N. M. da S. O., Soares, R. de S. C., Monteiro, E. L. T., Martins, S. C. R., Cavalcante, J. R., Grempel, R. G., & Neto, J. A. de O. (2016). Additive Manufacturing for Surgical Planning of Mandibular Fracture. Acta Stomatologica Croatica, 50(4), 348–353. https://doi.org/10.15644/asc50/4/8

BRUBAKER, C., FRECKER, T., NJOROGE, I., JENNINGS, G. K., ROSENTHAL, S., & ADAMS, D. (2017). Incorporation of Gold Nanoparticles for Enhanced Additive Manufacturing and 3D Printing Applications of Novel 'Smart' Materials. Structural Health Monitoring 2017, 0(shm). https://doi.org/10.12783/shm2017/14072

Budzik, G., Burek, J., Bazan, A., & Turek, P. (2016). Analysis of the Accuracy of Reconstructed Two Teeth Models Manufactured Using the 3DP and FDM Technologies. Strojniški Vestnik - Journal of Mechanical Engineering, 62(1). https://doi.org/10.5545/sv-jme.2015.2699

Cheng, Y. L., & Chen, S. J. (2006). Manufacturing of Cardiac Models Through Rapid Prototyping Technology for Surgery Planning. Materials Science Forum, 505–507, 1063–1068. https://doi.org/10.4028/www.scientific.net/MSF.505-507.1063

Chlebus, E., Kuźnicka, B., Kurzynowski, T., & Dybała, B. (2011). Microstructure and mechanical behaviour of Ti—6Al—7Nb alloy produced by selective laser melting. Materials Characterization, 62(5), 488–495. https://doi.org/10.1016/j.matchar.2011.03.006

Choi, A. H., Conway, R. C., Cazalbou, S., & Ben-Nissan, B. (2018). Maxillofacial bioceramics in tissue engineering: Production techniques, properties, and applications. In Fundamental Biomaterials: Ceramics (pp. 63–93). Elsevier. https://doi.org/10.1016/B978-0-08-102203-0.00003-2

Chougule, V. N., Mulay, A. V., & Ahuja, B. B. (2014). Development of patient specific implants for Minimum Invasive Spine Surgeries (MISS) from non-invasive imaging techniques by reverse engineering and additive manufacturing techniques. Procedia Engineering, 97, 212–219. https://doi.org/10.1016/j.proeng.2014.12.244

Clinkenbeard, R. E., Johnson, D. L., Parthasarathy, R., Altan, M. C., Tan, K.-H., Park, S.-M., & Crawford, R. H. (2002). Replication of Human Tracheobronchial Hollow Airway Models Using a Selective Laser Sintering Rapid Prototyping Technique. AIHAJ, 63(2), 141–150. https://doi.org/10.1202/0002-8894(2002)063<0141:ROHTHA>2.0.CO;2

Cooke, M. N., Fisher, J. P., Dean, D., Rimnac, C., & Mikos, A. G. (2003). Use of stereolithography to manufacture critical-sized 3D biodegradable scaffolds for bone ingrowth. Journal of Biomedical Materials Research, 64B(2), 65–69. https://doi.org/10.1002/jbm.b.10485

Cronskär, M., Bäckström, M., & Rännar, L. (2013). Production of customized hip stem prostheses – a comparison between conventional machining and electron beam melting (EBM). Rapid Prototyping Journal, 19(5), 365–372. https://doi.org/10.1108/RPJ-07-2011-0067

Cruz, F., & Coole, T. (2006). Additive fabrication of bioceramic/biopolymer bone implants. 95–96.

Dadbakhsh, S., Speirs, M., Van Humbeeck, J., & Kruth, J.-P. (2016). Laser additive manufacturing of bulk and porous shape-memory NiTi alloys: From processes to potential biomedical applications. MRS Bulletin, 41(10), 765–774. https://doi.org/10.1557/mrs.2016.209

Dahake, S. W., Kuthe, A. M., Chawla, J., & Mawale, M. B. (2017). Rapid prototyping assisted fabrication of customized surgical guides in mandibular distraction osteogenesis: a case report. Rapid Prototyping Journal, 23(3), 602–610. https://doi.org/10.1108/RPJ-09-2015-0129

Dahake, S. W., Kuthe, A. M., Mawale, M. B., & Bagde, A. D. (2016). Applications of medical rapid prototyping assisted customized surgical guides in complex surgeries. Rapid Prototyping Journal, 22(6), 934–946. https://doi.org/10.1108/RPJ-02-2015-0021

Daniel, S., & Eggbeer, D. (2016). A CAD and AM process for maxillofacial prostheses bar-clip retention. Rapid Prototyping Journal, 22(1), 170–177. https://doi.org/10.1108/RPJ-03-2014-0036

de Beer, N., & van der Merwe, A. (2013). Patient-specific intervertebral disc implants using rapid manufacturing technology. Rapid Prototyping Journal, 19(2), 126–139. https://doi.org/10.1108/13552541311302987

Dhariwala, B., Hunt, E., & Boland, T. (2004). Rapid Prototyping of Tissue-Engineering Constructs, Using Photopolymerizable Hydrogels and Stereolithography. Tissue Engineering, 10(9–10), 1316–1322. https://doi.org/10.1089/ten.2004.10.1316

Dobrzański, L. A. (2007). Archives of materials science and engineering international scientific journal published monthly as the organ of the Committee of Materials Science of the Polish Academy of Sciences. In Archives of Materials Science and Engineering (Issue Vol. 76, nr 2). International OCSCO World Press.

Duan, B., & Wang, M. (2011). Selective laser sintering and its application in biomedical engineering. MRS Bulletin, 36(12), 998–1005. https://doi.org/10.1557/mrs.2011.270

D'Urso, P. S., Effeney, D. J., Earwaker, W. J., Barker, T. M., Redmond, M. J., Thompson, R. G., & Tomlinson, F. H. (2000). Custom cranioplasty using stereolithography and acrylic. British Journal of Plastic Surgery, 53(3), 200–204. https://doi.org/10.1054/bjps.1999.3268

Edith Wiria, F., Fai Leong, K., & Kai Chua, C. (2010). Modeling of powder particle heat transfer process in selective laser sintering for fabricating tissue engineering scaffolds. Rapid Prototyping Journal, 16(6), 400–410. https://doi.org/10.1108/13552541011083317

Edith Wiria, F., Sudarmadji, N., Fai Leong, K., Kai Chua, C., Wei Chng, E., & Chai Chan, C. (2010). Selective laser sintering adaptation tools for cost effective fabrication of biomedical prototypes. Rapid Prototyping Journal, 16(2), 90–99. https://doi.org/10.1108/13552541011025816

Elahinia, M., Shayesteh Moghaddam, N., Taheri Andani, M., Amerinatanzi, A., Bimber, B. A., & Hamilton, R. F. (2016). Fabrication of NiTi through additive manufacturing: A review. Progress in Materials Science, 83, 630–663. https://doi.org/10.1016/J.PMATSCI.2016.08.001

El-Hajje, A., Kolos, E. C., Wang, J. K., Maleksaeedi, S., He, Z., Wiria, F. E., Choong, C., & Ruys, A. J. (2014). Physical and mechanical characterisation of 3D-printed porous titanium for biomedical applications. Journal of Materials Science: Materials in Medicine, 25(11), 2471–2480. https://doi.org/10.1007/s10856-014-5277-2

Espalin, D., Arcaute, K., Rodriguez, D., Medina, F., Posner, M., & Wicker, R. (2010). Fused deposition modeling of patient-specific polymethylmethacrylate implants. Rapid Prototyping Journal, 16(3), 164–173. https://doi.org/10.1108/13552541011034825

Facchini, L., Magalini, E., Robotti, P., & Molinari, A. (2009). Microstructure and mechanical properties of Ti-6Al-4V produced by electron beam melting of pre-alloyed powders. Rapid Prototyping Journal, 15(3), 171–178. https://doi.org/10.1108/13552540910960262

Falvo D'Urso Labate, G., Catapano, G., Vitale-Brovarone, C., & Baino, F. (2017). Quantifying the micro-architectural similarity of bioceramic scaffolds to bone. Ceramics International, 43(12), 9443–9450. https://doi.org/10.1016/j.ceramint.2017.04.121

Fatemi, A., Molaei, R., Sharifimehr, S., Shamsaei, N., & Phan, N. (2017). Torsional fatigue behavior of wrought and additive manufactured Ti-6Al-4V by powder bed fusion including surface finish effect. International Journal of Fatigue, 99, 187–201. https://doi.org/10.1016/J.IJFATIGUE.2017.03.002

Faure, S. P., Mercier, L., Didier, P., Roux, R., Coulon, J. F., Garel, S., Trenit, J., Buard, H., & Razan, F. (2012). Laser Sintering Process Analysis: Application to Chromium-Cobalt Alloys for Dental Prosthesis Production. Volume 4: Advanced Manufacturing Processes; Biomedical Engineering; Multiscale Mechanics of Biological Tissues; Sciences, Engineering and Education; Multiphysics; Emerging Technologies for Inspection, 9. https://doi.org/10.1115/ESDA2012-82108

Gallivanone, F., Interlenghi, M., Canervari, C., & Castiglioni, I. (2016). A fully automatic, threshold-based segmentation method for the estimation of the Metabolic Tumor Volume from PET images: validation on 3D printed anthropomorphic oncological lesions. Journal of Instrumentation, 11(01), C01022–C01022. https://doi.org/10.1088/1748-0221/11/01/C01022

Gauvin, R., Chen, Y.-C., Lee, J. W., Soman, P., Zorlutuna, P., Nichol, J. W., Bae, H., Chen, S., & Khademhosseini, A. (2012). Microfabrication of complex porous tissue engineering scaffolds using 3D projection stereolithography. Biomaterials, 33(15), 3824–3834. https://doi.org/10.1016/j.biomaterials.2012.01.048

Gebhardt, A., Schmidt, F.-M., Hötter, J.-S., Sokalla, W., & Sokalla, P. (2010). Additive Manufacturing by selective laser melting the realizer desktop machine and its application for the dental industry. Physics Procedia, 5, 543–549. https://doi.org/10.1016/j.phpro.2010.08.082

Gmeiner, R., & Deisinger, U. (2015). Additive manufacturing of bioactive glasses and silicate bioceramics. Researchgate.Net.

Goffard, R., & Sforza, T. (2013). Additive manufacturing of biocompatible ceramics. Search.Proquest.Com.

Gronet, P. M., Waskewicz, G. A., & Richardson, C. (2003). Preformed acrylic cranial implants using fused deposition modeling: A clinical report. The Journal of Prosthetic Dentistry, 90(5), 429–433. https://doi.org/10.1016/j.prosdent.2003.08.023

Hagedorn-Hansen, D., Oosthuizen, G. A., & Gerhold, T. (2016). RESOURCE-EFFICIENT PROCESS CHAINS TO MANUFACTURE PATIENT-SPECIFIC PROSTHETIC FINGERS. The South African Journal of Industrial Engineering, 27(1). https://doi.org/10.7166/27-1-1279

Hinderdael, M., Strantza, M., De Baere, D., Devesse, W., De Graeve, I., Terryn, H., & Guillaume, P. (2017). Fatigue Performance of Ti-6Al-4V Additively Manufactured Specimens with Integrated Capillaries of an Embedded Structural Health Monitoring System. Materials, 10(9), 993. https://doi.org/10.3390/ma10090993



Ho, D., Squelch, A., & Sun, Z. (2017). Modelling of aortic aneurysm and aortic dissection through 3D printing. Journal of Medical Radiation Sciences, 64(1), 10–17. https://doi.org/10.1002/jmrs.212

Höfer, R., & Hinrichs, K. (2009). Additives for the Manufacture and Processing of Polymers (pp. 97–145). Springer, Berlin, Heidelberg. https://doi.org/10.1007/698_2009_12

Hong, Y., Wu, M., Chen, G., Dai, Z., Zhang, Y., Chen, G., & Dong, X. (2016). 3D Printed Microfluidic Device with Microporous Mn2O3-Modified Screen Printed Electrode for Real-Time Determination of Heavy Metal Ions. ACS Applied Materials & Interfaces, 8(48), 32940–32947. https://doi.org/10.1021/acsami.6b10464

Hrabe, N., & Quinn, T. (2013a). Effects of processing on microstructure and mechanical properties of a titanium alloy (Ti–6Al–4V) fabricated using electron beam melting (EBM), part 1: Distance from build plate and part size. Materials Science and Engineering: A, 573, 264–270. https://doi.org/10.1016/J.MSEA.2013.02.064

Hrabe, N., & Quinn, T. (2013b). Effects of processing on microstructure and mechanical properties of a titanium alloy (Ti–6Al–4V) fabricated using electron beam melting (EBM), Part 2: Energy input, orientation, and location. Materials Science and Engineering: A, 573, 271–277. https://doi.org/10.1016/J.MSEA.2013.02.065

Huang, H., Xiang, C., Zeng, C., Ouyang, H., Wong, K. K. L., & Huang, W. (2015). Patient-specific geometrical modeling of orthopedic structures with high efficiency and accuracy for finite element modeling and 3D printing. Australasian Physical & Engineering Sciences in Medicine, 38(4), 743–753. https://doi.org/10.1007/s13246-015-0402-1

Husár, B., Hatzenbichler, M., Mironov, V., Liska, R., Stampfl, J., & Ovsianikov, A. (2014). Photopolymerization-based additive manufacturing for the development of 3D porous scaffolds. In Biomaterials for Bone Regeneration (pp. 149–201). Elsevier. https://doi.org/10.1533/9780857098104.2.149

Ionita, C. N., Mokin, M., Varble, N., Bednarek, D. R., Xiang, J., Snyder, K. V., Siddiqui, A. H., Levy, E. I., Meng, H., & Rudin, S. (2014). Challenges and limitations of patient-specific vascular phantom fabrication using 3D Polyjet printing (R. C. Molthen & J. B. Weaver, Eds.; p. 90380M). https://doi.org/10.1117/12.2042266

Jackson, A., Ray, L. A., Dangi, S., Ben-Zikri, Y. K., & Linte, C. A. (2017). 3D printing for orthopedic applications: from high resolution cone beam CT images to life size physical models (T. S. Cook & J. Zhang, Eds.; p. 101380T). https://doi.org/10.1117/12.2256181

Jardini, A. L., Larosa, M. A., Filho, R. M., Zavaglia, C. A. de C., Bernardes, L. F., Lambert, C. S., Calderoni, D. R., & Kharmandayan, P. (2014). Cranial reconstruction: 3D biomodel and custom-built implant created using additive manufacturing. Journal of Cranio-Maxillofacial Surgery, 42(8), 1877–1884. https://doi.org/10.1016/j.jcms.2014.07.006

Ji, Z., Yan, C., Yu, B., Wang, X., & Zhou, F. (2017). Multimaterials 3D Printing for Free Assembly Manufacturing of Magnetic Driving Soft Actuator. Advanced Materials Interfaces, 4(22), 1700629. https://doi.org/10.1002/admi.201700629

Jiménez, M., Romero, L., Domínguez, M., & Espinosa, M. M. (2015). Rapid prototyping model for the manufacturing by thermoforming of occlusal splints. Rapid Prototyping Journal, 21(1), 56–69. https://doi.org/10.1108/RPJ-11-2012-0101

Jungst, T., Smolan, W., Schacht, K., Scheibel, T., & Groll, J. (2016). Strategies and Molecular Design Criteria for 3D Printable Hydrogels. Chemical Reviews, 116(3), 1496–1539. https://doi.org/10.1021/acs.chemrev.5b00303

Koike, M., Martinez, K., Guo, L., Chahine, G., Kovacevic, R., & Okabe, T. (2011). Evaluation of titanium alloy fabricated using electron beam melting system for dental applications. Journal of Materials Processing Technology, 211(8), 1400–1408. https://doi.org/10.1016/j.jmatprotec.2011.03.013

Kong, Y. L., Gupta, M. K., Johnson, B. N., & McAlpine, M. C. (2016). 3D printed bionic nanodevices. Nano Today, 11(3), 330–350. https://doi.org/10.1016/J.NANTOD.2016.04.007

Koptioug, A., Rännar, L. E., Bäckström, M., & Klingvall, R. P. (2012). Electron Beam Melting: Moving from Macro- to Micro- and Nanoscale. Materials Science Forum, 706–709, 532–537. https://doi.org/10.4028/www.scientific.net/MSF.706-709.532

Koumoulos, E. P., Gkartzou, E., & Charitidis, C. A. (2017). Additive (nano)manufacturing perspectives: the use of nanofillers and tailored materials. Manufacturing Review, 4, 12. https://doi.org/10.1051/mfreview/2017012

Kruth, J. P., Wang, X., Laoui, T., & Froyen, L. (2003). Lasers and materials in selective laser sintering. Assembly Automation, 23(4), 357–371. https://doi.org/10.1108/01445150310698652

Kuk, M., Mitsouras, D., Dill, K. E., Rybicki, F. J., & Dwivedi, G. (2017). 3D Printing from Cardiac Computed Tomography for Procedural Planning. Current Cardiovascular Imaging Reports, 10(7), 21. https://doi.org/10.1007/s12410-017-9420-6

Kuo, C.-C., Chen, W.-H., Li, J.-F., & Zhu, Y.-J. (2018). Development of a flexible modeling base for additive manufacturing. The International Journal of Advanced Manufacturing Technology, 94(1–4), 1533–1541. https://doi.org/10.1007/s00170-017-1028-0

Lathers, S., & La Belle, J. (2016). Advanced Manufactured Fused Filament Fabrication 3D Printed Osseointegrated Prosthesis for a Transhumeral Amputation Using Taulman 680 FDA. 3D Printing and Additive Manufacturing, 3(3), 166–174. https://doi.org/10.1089/3dp.2016.0010

Lee, K.-W., Wang, S., Fox, B. C., Ritman, E. L., Yaszemski, M. J., & Lu, L. (2007). Poly(propylene fumarate) Bone Tissue Engineering Scaffold Fabrication Using Stereolithography: Effects of Resin Formulations and Laser Parameters. Biomacromolecules, 8(4), 1077–1084. https://doi.org/10.1021/bm060834v

Leonards, H., Engelhardt, S., Hoffmann, A., Pongratz, L., Schriever, S., Bläsius, J., Wehner, M., & Gillner, A. (2015). Advantages and drawbacks of Thiol-ene based resins for 3D-printing (H. Helvajian, A. Piqué, M. Wegener, & B. Gu, Eds.; Vol. 9353, p. 93530F). International Society for Optics and Photonics. https://doi.org/10.1117/12.2081169

Leuders, S., Thöne, M., Riemer, A., Niendorf, T., Tröster, T., Richard, H. A., & Maier, H. J. (2013). On the mechanical behaviour of titanium alloy TiAl6V4 manufactured by selective laser melting: Fatigue resistance and crack growth performance. International Journal of Fatigue, 48, 300–307. https://doi.org/10.1016/J.IJFATIGUE.2012.11.011

Li, X., Wang, C., Zhang, W., & Li, Y. (2009). Fabrication and characterization of porous Ti6Al4V parts for biomedical applications using electron beam melting process. Materials Letters, 63(3–4), 403–405. https://doi.org/10.1016/j.matlet.2008.10.065

Li, X., Wang, Y., Zhao, Y., Liu, J., Xiao, S., & Mao, K. (2017). Multilevel 3D Printing Implant for Reconstructing Cervical Spine with Metastatic Papillary Thyroid Carcinoma. SPINE, 42(22), E1326–E1330. https://doi.org/10.1097/BRS.0000000000002229

Liravi, F., & Toyserkani, E. (2018). A hybrid additive manufacturing method for the fabrication of silicone bio-structures: 3D printing optimization and surface characterization. Materials & Design, 138, 46–61. https://doi.org/10.1016/J.MATDES.2017.10.051

Liu, Q., Leu, M. C., & Schmitt, S. M. (2006). Rapid prototyping in dentistry: technology and application. The International Journal of Advanced Manufacturing Technology, 29(3–4), 317–335. https://doi.org/10.1007/s00170-005-2523-2

Lopes, G., Miranda, R. M., Quintino, L., Rodrigues, J. P. (2007). Additive manufacturing of Ti-6Al-4V based components with high power fiber lasers. Virtual and Rapid Manufacturing, 369–374.

Lueders, C., Jastram, B., Hetzer, R., & Schwandt, H. (2014). Rapid manufacturing techniques for the tissue engineering of human heart valves. European Journal of Cardio-Thoracic Surgery, 46(4), 593–601. https://doi.org/10.1093/ejcts/ezt510

Lusquiños, F., del Val, J., Arias-González, F., Comesaña, R., Quintero, F., Riveiro, A., Boutinguiza, M., Jones, J. R., Hill, R. G., & Pou, J. (2014). Bioceramic 3D Implants Produced by Laser Assisted Additive Manufacturing. Physics Procedia, 56, 309–316. https://doi.org/10.1016/j.phpro.2014.08.176

M Zanetti, E., Aldieri, A., Terzini, M., Calì, M., Franceschini, G., & Bignardi, C. (2017). ADDITIVELY MANUFACTURED CUSTOM LOAD-BEARING IMPLANTABLE DEVICES. Australasian Medical Journal, 10(08). https://doi.org/10.21767/AMJ.2017.3093

MacBarb, R. F., Lindsey, D. P., Bahney, C. S., Woods, S. A., Wolfe, M. L., & Yerby, S. A. (2017). Fortifying the Bone-Implant Interface Part 1: An In Vitro Evaluation of 3D-Printed and TPS Porous Surfaces. International Journal of Spine Surgery, 11, 15. https://doi.org/10.14444/4015

McCullough, E. J., & Yadavalli, V. K. (2013). Surface modification of fused deposition modeling ABS to enable rapid prototyping of biomedical microdevices. Journal of Materials Processing Technology, 213(6), 947–954. https://doi.org/10.1016/j.jmatprotec.2012.12.015

Melchels, F. P. W., Feijen, J., & Grijpma, D. W. (2009). A poly(d,l-lactide) resin for the preparation of tissue engineering scaffolds by stereolithography. Biomaterials, 30(23–24), 3801–3809. https://doi.org/10.1016/j.biomaterials.2009.03.055

Melchels, F. P. W., Feijen, J., & Grijpma, D. W. (2010). A review on stereolithography and its applications in biomedical engineering. Biomaterials, 31(24), 6121–6130. https://doi.org/10.1016/j.biomaterials.2010.04.050

Misra, S. K., Ostadhossein, F., Babu, R., Kus, J., Tankasala, D., Sutrisno, A., Walsh, K. A., Bromfield, C. R., & Pan, D. (2017). 3D-Printed Multidrug-Eluting Stent from Graphene-Nanoplatelet-Doped Biodegradable Polymer Composite. Advanced Healthcare Materials, 6(11), 1700008. https://doi.org/10.1002/adhm.201700008

Mohamed, O. A., Masood, S. H., & Bhowmik, J. L. (2015). Optimization of fused deposition modeling process parameters: a review of current research and future prospects. Advances in Manufacturing, 3(1), 42–53. https://doi.org/10.1007/s40436-014-0097-7

Msallem, B., Beiglboeck, F., Honigmann, P., Jaquiéry, C., & Thieringer, F. (2017). Craniofacial Reconstruction by a Cost-Efficient Template-Based Process Using 3D Printing. Plastic and Reconstructive Surgery—Global Open, 5(11), e1582. https://doi.org/10.1097/GOX.0000000000001582

Mullen, L., Stamp, R. C., Brooks, W. K., Jones, E., & Sutcliffe, C. J. (2009). Selective Laser Melting: A regular unit cell approach for the manufacture of porous, titanium, bone in-growth constructs, suitable for orthopedic applications. Journal of Biomedical Materials Research Part B: Applied Biomaterials, 89B(2), 325–334. https://doi.org/10.1002/jbm.b.31219

Murr, L. E., Amato, K. N., Li, S. J., Tian, Y. X., Cheng, X. Y., Gaytan, S. M., Martinez, E., Shindo, P. W., Medina, F., & Wicker, R. B. (2011). Microstructure and mechanical properties of open-cellular biomaterials prototypes for total knee replacement implants fabricated by electron beam melting. Journal of the Mechanical Behavior of Biomedical Materials, 4(7), 1396–1411. https://doi.org/10.1016/j.jmbbm.2011.05.010

Murr, L. E., Gaytan, S. M., Martinez, E., Medina, F., & Wicker, R. B. (2012). Next Generation Orthopaedic Implants by Additive Manufacturing Using Electron Beam Melting. International Journal of Biomaterials, 2012, 1–14. https://doi.org/10.1155/2012/245727

Nabiyouni, M., Brückner, T., Zhou, H., Gbureck, U., & Bhaduri, S. B. (2018). Magnesium-based bioceramics in orthopedic applications. Acta Biomaterialia, 66, 23–43. https://doi.org/10.1016/j.actbio.2017.11.033

Nakano, T., & Ishimoto, T. (2015). Powder-based Additive Manufacturing for Development of Tailor-made Implants for Orthopedic Applications. KONA Powder and Particle Journal, 32(0), 75–84. https://doi.org/10.14356/kona.2015015

Nayar, S., Bhuminathan, S., & Bhat, W. (2015). Rapid prototyping and stereolithography in dentistry. Journal of Pharmacy and Bioallied Sciences, 7(5), 218. https://doi.org/10.4103/0975-7406.155913

Nocerino, E., Remondino, F., Uccheddu, F., Gallo, M., & Gerosa, G. (2016). 3D MODELLING AND RAPID PROTOTYPING FOR CARDIOVASCULAR SURGICAL PLANNING – TWO CASE STUDIES. The International Archives of the Photogrammetry, Remote Sensing and Spatial Information Sciences. https://doi.org/10.5194/isprsarchives-XLI-B5-887-2016

Ogden, K., Ordway, N., Diallo, D., Tillapaugh-Fay, G., & Aslan, C. (2014). Dimensional accuracy of 3D printed vertebra (Z. R. Yaniv & D. R. Holmes, Eds.; p. 903629). https://doi.org/10.1117/12.2043489

O'Hara, R. P., Chand, A., Vidiyala, S., Arechavala, S. M., Mitsouras, D., Rudin, S., & Ionita, C. N. (2016). Advanced 3D mesh manipulation in stereolithographic files and post-print processing for the manufacturing of patient-specific vascular flow phantoms (J. Zhang & T. S. Cook, Eds.; p. 978909). https://doi.org/10.1117/12.2217036

Opolski, A. C., Erbano, B. O., Schio, N. A., de Salles Graça, Y. L. S., Guarinello, G. G., de Oliveira, P. M., Leal, A. G., Foggiatto, J. A., & Kubrusly, L. F. (2014). Experimental Three-Dimensional Biomodel of Complex Aortic Aneurysms by Rapid Prototyping Technology. 3D Printing and Additive Manufacturing, 1(2), 88–94. https://doi.org/10.1089/3dp.2013.0009

Pan, Y., Patil, A., Guo, P., & Zhou, C. (2017). A novel projection based electro-stereolithography (PES) process for production of 3D polymer-particle composite objects. Rapid Prototyping Journal, 23(2), 236–245. https://doi.org/10.1108/RPJ-02-2016-0030

Peel, S., Bhatia, S., Eggbeer, D., Morris, D. S., & Hayhurst, C. (2017). Evolution of design considerations in complex craniofacial reconstruction using patient-specific implants. Proceedings of the Institution of Mechanical Engineers, Part H: Journal of Engineering in Medicine, 231(6), 509–524. https://doi.org/10.1177/0954411916681346

Pekkanen, A. M., Mondschein, R. J., Williams, C. B., & Long, T. E. (2017). 3D Printing Polymers with Supramolecular Functionality for Biological Applications. Biomacromolecules, 18(9), 2669–2687. https://doi.org/10.1021/acs.biomac.7b00671

Petcu, E. B. (2017). 3D Bio-Printing: an Introduction to a New Approach for Cancer Patients at the Interface of Art and Medicine. Leonardo, 50(2), 195–196. https://doi.org/10.1162/LEON_a_01418

Poh, P. S. P., Chhaya, M. P., Wunner, F. M., De-Juan-Pardo, E. M., Schilling, A. F., Schantz, J.-T., van Griensven, M., & Hutmacher, D. W. (2016). Polylactides in additive biomanufacturing. Advanced Drug Delivery Reviews, 107, 228–246. https://doi.org/10.1016/j.addr.2016.07.006

Popescu, D., Laptoiu, D., Hadar, A., Ilie, C., & Parvu, C. (2015). How to design and additive manufacture individualized surgical guides for hand osteotomy. 2015 E-Health and Bioengineering Conference (EHB), 1–4. https://doi.org/10.1109/EHB.2015.7391609

Popescu, D., Lăptoiu, D., Marinescu, R., Hadar, A., & Botezatu, I. (2017). Advanced Engineering in Orthopedic Surgery Applications. Key Engineering Materials, 752, 99–104. https://doi.org/10.4028/www.scientific.net/KEM.752.99

Popovich, A., Sufiiarov, V., Polozov, I., Borisov, E., & Masaylo, D. (2016). Additive manufacturing of individual implants from titanium alloy. 1504–1508.

Radosh, A., Kuczko, W., Wichniarek, R., & Górski, F. (2017). Prototyping of Cosmetic Prosthesis of Upper Limb Using Additive Manufacturing Technologies. Advances in Science and Technology Research Journal, 11(3), 102–108. https://doi.org/10.12913/22998624/70995

Rahmati, S., Abbaszadeh, F., & Farahmand, F. (2012). An improved methodology for design of custom-made hip prostheses to be fabricated using additive manufacturing technologies. Rapid Prototyping Journal, 18(5), 389–400. https://doi.org/10.1108/13552541211250382

Ramakrishnaiah, R., Al kheraif, A. A., Mohammad, A., Divakar, D. D., Kotha, S. B., Celur, S. L., Hashem, M. I., Vallittu, P. K., & Rehman, I. U. (2017). Preliminary fabrication and characterization of electron beam melted Ti–6Al–4V customized dental implant. Saudi Journal of Biological Sciences, 24(4), 787–796. https://doi.org/10.1016/j.sjbs.2016.05.001

Ramasamy, M., & Varadan, V. K. (2016). 3D printing of nano- and micro-structures (V. K. Varadan, Ed.; Vol. 9802, p. 98020H). International Society for Optics and Photonics. https://doi.org/10.1117/12.2224069

Rimell, J. T., & Marquis, P. M. (2000). Selective laser sintering of ultra high molecular weight polyethylene for clinical applications. Journal of Biomedical Materials Research, 53(4), 414–420. https://doi.org/10.1002/1097-4636(2000)53:4<414::AID-JBM16>3.0.CO;2-M

Rogers, B., Bosker, G. W., Crawford, R. H., Faustini, M. C., Neptune, R. R., Walden, G., & Gitter, A. J. (2007). Advanced Trans-Tibial Socket Fabrication Using Selective Laser Sintering. Prosthetics and Orthotics International, 31(1), 88–100. https://doi.org/10.1080/03093640600983923

Ryan, J. R., Almefty, K. K., Nakaji, P., & Frakes, D. H. (2016). Cerebral Aneurysm Clipping Surgery Simulation Using Patient-Specific 3D Printing and Silicone Casting. World Neurosurgery, 88, 175–181. https://doi.org/10.1016/j.wneu.2015.12.102

Sahoo, S. (2014). Microstructure simulation of Ti-6Al-4V biomaterial produced by electron beam additive manufacturing process. International Journal of Nano and Biomaterials, 5(4), 228. https://doi.org/10.1504/IJNBM.2014.069811

Sahoo, S., & Chou, K. (2016). Phase-field simulation of microstructure evolution of Ti–6Al–4V in electron beam additive manufacturing process. Additive Manufacturing, 9, 14–24. https://doi.org/10.1016/J.ADDMA.2015.12.005

Sankar, S., Paulose, J., & Thomas, N. (2017). 3D Printed Quick Healing Cast: The Exoskeletal Immobilizer. Volume 14: Emerging Technologies; Materials: Genetics to Structures; Safety Engineering and Risk Analysis, V014T07A008. https://doi.org/10.1115/IMECE2017-71252

Schantz, J.-T., Brandwood, A., Hutmacher, D. W., Khor, H. L., & Bittner, K. (2005). Osteogenic differentiation of mesenchymal progenitor cells in computer designed fibrin-polymer-ceramic scaffolds manufactured by fused deposition modeling. Journal of Materials Science: Materials in Medicine, 16(9), 807–819. https://doi.org/10.1007/s10856-005-3584-3

Schmidt, M., Pohle, D., & Rechtenwald, T. (2007). Selective Laser Sintering of PEEK. CIRP Annals, 56(1), 205–208. https://doi.org/10.1016/j.cirp.2007.05.097

Schrank, E. S., Hitch, L., Wallace, K., Moore, R., & Stanhope, S. J. (2013). Assessment of a Virtual Functional Prototyping Process for the Rapid Manufacture of Passive-Dynamic Ankle-Foot Orthoses. Journal of Biomechanical Engineering, 135(10), 101011. https://doi.org/10.1115/1.4024825

Shin, J., Sandhu, R. S., & Shih, G. (2017). Imaging Properties of 3D Printed Materials: Multi-Energy CT of Filament Polymers. Journal of Digital Imaging, 30(5), 572–575. https://doi.org/10.1007/s10278-017-9954-9

Shishkovsky, I. V., Volova, L. T., Kuznetsov, M. V., Morozov, Yu. G., & Parkin, I. P. (2008). Porous biocompatible implants and tissue scaffolds synthesized by selective laser sintering from Ti and NiTi. Journal of Materials Chemistry, 18(12), 1309. https://doi.org/10.1039/b715313a

Short, D. B., Volk, D., Badger, P. D., Melzer, J., Salerno, P., & Sirinterlikci, A. (2014). 3D Printing (Rapid Prototyping) Photopolymers: An Emerging Source of Antimony to the Environment. 3D Printing and Additive Manufacturing, 1(1), 24–33. https://doi.org/10.1089/3dp.2013.0001

Sidambe, A. (2014). Biocompatibility of Advanced Manufactured Titanium Implants—A Review. Materials, 7(12), 8168–8188. https://doi.org/10.3390/ma7128168

Sindhu, V., & Soundarapandian, S. (2017). Additive Manufacturing Fixture Box for Bone Measurement. Procedia Engineering, 184, 1–9. https://doi.org/10.1016/j.proeng.2017.04.063

Singare, S., Yaxiong, L., Dichen, L., Bingheng, L., Sanhu, H., & Gang, L. (2006). Fabrication of customised maxillo-facial prosthesis using computer-aided design and rapid prototyping techniques. Rapid Prototyping Journal, 12(4), 206–213. https://doi.org/10.1108/13552540610682714

Sljivic, M., Stanojevic, M., Djurdjevic, D., Grujovic, N., & Pavlovic, A. (2016). Implementation of FEM and rapid prototyping in maxillofacial surgery. FME Transaction, 44(4), 422–429. https://doi.org/10.5937/fmet1604422S

Smith, M. L., McGuinness, J., O'Reilly, M. K., Nolke, L., Murray, J. G., & Jones, J. F. X. (2017). The role of 3D printing in preoperative planning for heart transplantation in complex congenital heart disease. Irish Journal of Medical Science (1971-), 186(3), 753–756. https://doi.org/10.1007/s11845-017-1564-5

Soon, D. S. C., Chae, M. P., Pilgrim, C. H. C., Rozen, W. M., Spychal, R. T., & Hunter-Smith, D. J. (2016). 3D haptic modelling for preoperative planning of hepatic resection: A systematic review. Annals of Medicine and Surgery (2012), 10, 1–7. https://doi.org/10.1016/j.amsu.2016.07.002

Spallek, J., & Krause, D. (2016). Process Types of Customisation and Personalisation in Design for Additive Manufacturing Applied to Vascular Models. Procedia CIRP, 50, 281–286. https://doi.org/10.1016/j.procir.2016.05.022

Srivatsan, T., & Sudarshan, T. (2015). Additive manufacturing: innovations, advances, and applications.

Stieghorst, J., Majaura, D., Wevering, H., & Doll, T. (2016). Toward 3D Printing of Medical Implants: Reduced Lateral Droplet Spreading of Silicone Rubber under Intense IR Curing. ACS Applied Materials & Interfaces, 8(12), 8239–8246. https://doi.org/10.1021/acsami.5b12728

Strano, G., Hao, L., Everson, R. M., & Evans, K. E. (2013). Surface roughness analysis, modelling and prediction in selective laser melting. Journal of Materials Processing Technology, 213(4), 589–597. https://doi.org/10.1016/j.jmatprotec.2012.11.011

Sugioka, K., & Cheng, Y. (2014). Femtosecond laser three-dimensional micro- and nanofabrication. Applied Physics Reviews, 1(4), 041303. https://doi.org/10.1063/1.4904320

Suwanprateeb, J., Thammarakcharoen, F., & Suvannapruk, W. (2014). Preparation and Characterization of 3D Printed Porous Polyethylene for Medical Applications by Novel Wet Salt Bed Technique. J. Sci. Chiang Mai J. Sci, 41(411), 200–212.

Thomas, D. J., Azmi, M. A. B. M., & Tehrani, Z. (2014). 3D additive manufacture of oral and maxillofacial surgical models for preoperative planning. The International Journal of Advanced Manufacturing Technology, 71(9–12), 1643–1651. https://doi.org/10.1007/s00170-013-5587-4

Tie, Y., Ma, R., Ye, M., Wang, D., & Wang, C. (2006). Rapid prototyping fabrication and finite element evaluation of the three-dimensional medical pelvic model. The International Journal of Advanced Manufacturing Technology, 28(3–4), 302–306. https://doi.org/10.1007/s00170-004-2377-z

Tröger, C., Bens, A. T., Bermes, G., Klemmer, R., Lenz, J., & Irsen, S. (2008). Ageing of acrylate-based resins for stereolithography: thermal and humidity ageing behaviour studies. Rapid Prototyping Journal, 14(5), 305–317. https://doi.org/10.1108/13552540810907983

Vaezi, M., & Yang, S. (2015). Extrusion-based additive manufacturing of PEEK for biomedical applications. Virtual and Physical Prototyping, 10(3), 123–135. https://doi.org/10.1080/17452759.2015.1097053

Vandenbroucke, B., & Kruth, J. (2007). Selective laser melting of biocompatible metals for rapid manufacturing of medical parts. Rapid Prototyping Journal, 13(4), 196–203. https://doi.org/10.1108/13552540710776142

Vitali, A., Regazzoni, D., Rizzi, C., & Colombo, G. (2017). Design and Additive Manufacturing of Lower Limb Prosthetic Socket. Volume 11: Systems, Design, and Complexity, V011T15A021. https://doi.org/10.1115/IMECE2017-71494

Walker, J. M., Bodamer, E., Krebs, O., Luo, Y., Kleinfehn, A., Becker, M. L., & Dean, D. (2017). Effect of Chemical and Physical Properties on the In Vitro Degradation of 3D Printed High Resolution Poly(propylene fumarate) Scaffolds. Biomacromolecules, 18(4), 1419–1425. https://doi.org/10.1021/acs.biomac.7b00146

Wang, M., Lin, X., & Huang, W. (2016). Laser additive manufacture of titanium alloys. Materials Technology, 1–8. https://doi.org/10.1179/1753555715Y.0000000079

Wei, Q., Li, S., Han, C., Li, W., Cheng, L., Hao, L., & Shi, Y. (2015). Selective laser melting of stainless-steel/nano-hydroxyapatite composites for medical applications: Microstructure, element distribution, crack and mechanical properties. Journal of Materials Processing Technology, 222, 444–453. https://doi.org/10.1016/j.jmatprotec.2015.02.010

Williams, J. M., Adewunmi, A., Schek, R. M., Flanagan, C. L., Krebsbach, P. H., Feinberg, S. E., Hollister, S. J., & Das, S. (2005). Bone tissue engineering using polycaprolactone scaffolds fabricated via selective laser sintering. Biomaterials, 26(23), 4817–4827. https://doi.org/10.1016/j.biomaterials.2004.11.057

Winder, J., & Bibb, R. (2005). Medical Rapid Prototyping Technologies: State of the Art and Current Limitations for Application in Oral and Maxillofacial Surgery. Journal of Oral and Maxillofacial Surgery, 63(7), 1006–1015. https://doi.org/10.1016/j.joms.2005.03.016

Winder, J., Cooke, R. S., Gray, J., Fannin, T., & Fegan, T. (1999). Medical rapid prototyping and 3D CT in the manufacture of custom made cranial titanium plates. Journal of Medical Engineering & Technology, 23(1), 26–28. https://doi.org/10.1080/030919099294401

Witowski, J. S., Coles-Black, J., Zuzak, T. Z., Pędziwiatr, M., Chuen, J., Major, P., & Budzyński, A. (2017). 3D Printing in Liver Surgery: A Systematic Review. Telemedicine and E-Health, 23(12), 943–947. https://doi.org/10.1089/tmj.2017.0049

Witowski, J. S., Pędziwiatr, M., Major, P., & Budzyński, A. (2017). Cost-effective, personalized, 3D-printed liver model for preoperative planning before laparoscopic liver hemihepatectomy for colorectal cancer metastases. International Journal of Computer Assisted Radiology and Surgery, 12(12), 2047–2054. https://doi.org/10.1007/s11548-017-1527-3

Wong, K. C. (2016). 3D-printed patient-specific applications in orthopedics. Orthopedic Research and Reviews, Volume 8, 57–66. https://doi.org/10.2147/ORR.S99614

Wu, W., Qin, X., Chen, Y., Wang, W., & Rosen, D. W. (2010). Employing Rapid Prototyping biomedical model to assist the surgical planning of defect mandibular reconstruction. 2010 3rd International Conference on Biomedical Engineering and Informatics, 1863–1866. https://doi.org/10.1109/BMEI.2010.5639569

Wu, W. Z., Geng, P., Zhao, J., Zhang, Y., Rosen, D. W., & Zhang, H. B. (2014). Manufacture and thermal deformation analysis of semicrystalline polymer polyether ether ketone by 3D printing. Materials Research Innovations, 18(sup5), S5-12-S5-16. https://doi.org/10.1179/1432891714Z.000000000898

Xu, N., Ye, X., Wei, D., Zhong, J., Chen, Y., Xu, G., & He, D. (2014). 3D Artificial Bones for Bone Repair Prepared by Computed Tomography-Guided Fused Deposition Modeling for Bone Repair. ACS Applied Materials & Interfaces, 6(17), 14952–14963. https://doi.org/10.1021/am502716t

Yang, Y., Lu, J., Luo, Z., & Wang, D. (2012). Accuracy and density optimization in directly fabricating customized orthodontic production by selective laser melting. Rapid Prototyping Journal, 18(6), 482–489. https://doi.org/10.1108/13552541211272027

Yves-Christian, H., Jan, W., Wilhelm, M., Konrad, W., & Reinhart, P. (2010). High value manufacturing Net shaped high performance oxide ceramic parts by selective laser melting. Physics Procedia. 5, 587–594. https://doi.org/https://doi.org/10.1016/J.PHPRO.2010.08.086 ad

Zein, I., Hutmacher, D. W., Tan, K. C., & Teoh, S. H. (2002). Fused deposition modeling of novel scaffold architectures for tissue engineering applications. Biomaterials, 23(4), 1169–1185. https://doi.org/10.1016/S0142-9612(01)00232-0

Zhai, Y., Galarraga, H., & Lados, D. A. (2016). Microstructure, static properties, and fatigue crack growth mechanisms in Ti-6Al-4V fabricated by additive manufacturing: LENS and EBM. Engineering Failure Analysis, 69, 3–14. https://doi.org/10.1016/J.ENGFAILANAL.2016.05.036

Zhang, L. C., Klemm, D., Eckert, J., Hao, Y. L., & Sercombe, T. B. (2011). Manufacture by selective laser melting and mechanical behavior of a biomedical Ti–24Nb–4Zr–8Sn alloy. Scripta Materialia, 65(1), 21–24. https://doi.org/10.1016/j.scriptamat.2011.03.024

Zuniga, J., Katsavelis, D., Peck, J., Stollberg, J., Petrykowski, M., Carson, A., & Fernandez, C. (2015). Cyborg beast: a low-cost 3d-printed prosthetic hand for children with upper-limb differences. BMC Research Notes, 8(1), 10. https://doi.org/10.1186/s13104-015-0971-9

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
