# Peer review of "The Impact of Additive Manufacturing on Supply Chain Management from a System Dynamics Model—Scenario: Traditional, Centralized, and Distributed Supply Chain"

_processes, doi:10.3390/pr10122489_

Round 1

Reviewer 1 Report

I hope these comments will contribute to improve the quality of this paper:

1. The Introduction should be improved following the authors guidelines (https://www.mdpi.com/authors/layout#_bookmark0).

2. Authors must cite all references in the text. For example, all the cited references in Table 2 must be in Appendix A.

3. A Literature Review section should be included. For example, from this review, it would be possible to identify what are the problem and research question conducting this research, and how this research contributes to answer this question.

4. Figures must be clear, as some of them are difficult to read (e.g., Figures 19, 20, 21, and so on).

5. Should Table 1 (Line 249) be Table 3?

6. Line 664, Campuzano-Bolarín instead of Campuzano-bolarín.

7. Is there an Appendix B (Line 822).

Reviewer 2 Report

In this study, the authors studied the impact of additive manufacturing on supply chain management by comparing supply chain with traditional and additive manufacturing using a system dynamics model. The paper and the details are interesting and I propose this paper for publication after addressing the following issues and suggestions:

1.  1. I suggest authors to add “Discussion” section to discuss meaning, significance and implication of the results they obtained.

2.      2. There should not be discussion in “Materials and Methods” section. For instance, the first paragraph of M&M can be moved to the discussion part.

3.    3.   Conclusion part should be short and concise. Only key results and their significance should be presented in this section.

4.   4.    In line 37, the abbreviation of “System Dynamics” as SD is directly written, which should be mentioned first in line 31 and only abbreviation should be used in other places.

Best wishes in your work

Author Response

Please, see attachment

Reviewer 3 Report

This paper describes the impact of additive manufacturing on supply chain management from a system dynamics model. The language of the paper is good, and its structure is well organized. Although it is an interesting approach, there are some issues that the authors have to address:

1.       The abstract must describe better the results of the analysis and illustrate the research contribution of the present study.

2.       Section 2, could be extended to better analyze the

3.       Table 9 & 10 must be resized and adjusted to the journal’s format

4.       In Figure 14, remove the underline

5.       The Figures 19-22 must be of higher analysis in order to be more readable

6.       All the references must conform to the journal’s format

7. The conclusions section must be rewritten in order to include more information about the results of this study, and better describe the contribution to the body of knowledge, instead of repeating the three scenarios descriptions. In addition, a discussion sub-section could be inserted to analyze why the AM performance appears significantly better in conditions of lower demand
